# Mechanical signaling through membrane tension induces somal translocation during neuronal migration

Takunori Minegishi [ID] [1][✉], Honami Hasebe[1], Tomoya Aoyama[2], Keiji Naruse[3], Yasufumi Takahashi[2,4,5] & Naoyuki Inagaki [ID] [1][✉]

## Abstract

Neurons migrate in a saltatory manner by repeating two distinct steps: extension of the leading process and translocation of the cell body. The former step is critical for determining the migratory route in response to extracellular guidance cues. In the latter step, neurons must generate robust forces that translocate the bulky soma against mechanical barriers of the surrounding three-dimensional environment. However, the link between the leading process extension and subsequent somal translocation remains unknown. By using the membrane tension sensor Flipper-TR and scanning ion conductance microscopy, we show that leading process extension increases plasma membrane tension. The tension elevation activated the mechanosensitive ion channel Tmem63b and triggered $Ca^{2+}$ influx, leading to actomyosin activation at the rear of the cell. Blockade of this signaling pathway disturbed somal translocation, thereby inhibiting neuronal migration in three-dimensional environments. These data suggest that mechanical signaling through plasma membrane tension and mechano-channels links the leading process extension to somal translocation, allowing rapid and saltatory neuronal migration.

**Keywords** Neuronal Migration; Plasma Membrane Tension; Mechanosensing; Mechanosensitive Ion Channel; Myosin II
**Subject Categories** Cell Adhesion, Polarity & Cytoskeleton; Neuroscience; Signal Transduction

## Introduction

Neuronal migration is essential for brain development and neural network formation (Ross and Walsh, 2001; Hatten, 2002; Rakic, 2009; Saito et al, 2019; Nakajima et al, 2021). As a characteristic feature of neuronal migration, neurons navigate in a saltatory manner by repeating two steps (Edmondson and Hatten, 1987; O'Rourke et al, 1992). Their migration is initiated by the extension of the leading process, which determines the migratory route in response to extracellular guidance cues (O'Rourke et al, 1992; Marin et al, 2010). This is followed by the translocation of the soma toward the leading process. Because the soma is the largest part of the migrating neuron, it must generate robust forces to translocate against the mechanical barriers of the surrounding three-dimensional (3D) environment (Shu et al, 2004; Schaar and McConnell, 2005; Solecki et al, 2009; Zhang et al, 2009; He et al, 2010; Martini and Valdeolmillos, 2010).

A previous study reported that the leading process extension of olfactory interneurons is triggered by the accumulation of shootin1b at the growth cone. This accumulation generates the traction forces necessary for the process extension (Minegishi et al, 2018). On the other hand, somal translocation of cerebellar granule cells correlates with a transient increase in intracellular $Ca^{2+}$ concentration ($Ca^{2+}$ transient) (Komuro and Rakic, 1996). In medial ganglionic eminence cells, $Ca^{2+}$ transient induces actomyosin contraction at the rear of the cell, generating forces that drive somal translocation (Martini and Valdeolmillos, 2010). Importantly, somal translocation is initiated when the length of the leading process reaches a certain threshold (Wichterle et al, 1997; Schaar and McConnell, 2005), suggesting that the leading process extension is the key step in triggering somal translocation. However, the signaling pathway that links the leading process extension and somal translocation remains unknown.

In this study, we performed live imaging of plasma membrane tension in olfactory interneurons and found that plasma membrane tension increases during the leading process extension. The tension elevation activated the mechanosensitive ion channel Tmem63b (Murthy et al, 2018) and triggered $Ca^{2+}$ signaling, which in turn activated myosin II at the rear of the cell. Blockade of this signaling pathway by Tmem63b knockdown disturbed somal translocation and neuronal migration in 3D environments. These findings suggest that the tension-mediated signaling through mechano-channels links the extension of leading processes to the generation of robust forces for somal translocation, enabling efficient and saltatory 3D neuronal migration.

[1]Laboratory of Systems Neurobiology and Medicine, Division of Biological Science, Nara Institute of Science and Technology, Ikoma, Nara 630-0192, Japan. [2]WPI Nano Life Science Institute, Kanazawa University, Kanazawa, Ishikawa 920-1192, Japan. [3]Department of Cardiovascular Physiology, Graduate School of Medicine, Dentistry and Pharmaceutical Sciences, Okayama University, Kita-ku, Okayama 700-8558, Japan. [4]Department of Electronics, Graduate School of Engineering, Nagoya University, Nagoya, Aichi 464-8601, Japan. [5]Research Institute for Quantum and Chemical Innovation, Institutes of Innovation for Future Society, Nagoya University, Nagoya, Aichi 464-8601, Japan. [✉]E-mail: t-minegishi@bs.naist.jp; ninagaki@bs.naist.jp

# Results

## Leading process extension increases plasma membrane tension

A previous study using chick sensory neurons reported that the growth cone at the tip of neurites pulls their neurite during neurite extension (Lamoureux et al, 1989). In addition, recent studies reported that actin-based membrane protrusion leads to a global increase in plasma membrane tension (Houk et al, 2012; Tsujita et al, 2015; De Belly et al, 2023). To identify the signaling events triggered by leading process extension, we focused on the mechanical properties of migrating neurons. We first analyzed the membrane tension of migrating mouse olfactory interneurons in 3D Matrigel by measuring the fluorescence lifetime of the plasma membrane tension-sensitive probe Flipper-TR (Colom et al, 2018); a longer fluorescence lifetime of the probe indicates higher membrane tension. The plasma membrane tension was relatively homogeneous throughout the leading process shaft and soma (Fig. 1A; Appendix Fig. S1). Analyses of neurons bearing leading processes of different lengths revealed a significant positive correlation between the process length and plasma membrane tension in the soma and leading process shaft (Fig. 1A,B).

Since it has been reported that the fluorescence lifetime of Flipper-TR is affected by the order of the lipid membrane (Colom et al, 2018), we further investigated whether the leading process extension increases the plasma membrane tension using a different method. We analyzed the membrane tension with scanning ion conductance microscopy (SICM) (Rheinlaender and Schaffer, 2013) on 2D laminin substrate. SICM has been used to monitor plasma membrane tension through cell stiffness measurements (see Methods) (Rheinlaender and Schaffer, 2013; Bednarska et al, 2020). Consistent with the above data, the stiffness was relatively homogeneous across the leading process shaft and soma (Fig. 1C). As the actin cortex underlying the plasma membrane may affect the membrane stiffness (Rheinlaender and Schaffer, 2013), we measured the membrane stiffness at the same proximal regions of the leading process (Fig. 1D). By monitoring the change in stiffness of the same regions, we minimized the possible influence of differences in the local actin cortex. The stiffness of the membrane in the proximal region of the leading process increased as the process extended and decreased as it shortened (Fig. 1D). This relationship was confirmed by the significant positive correlation between the increased stiffness and the increase in the leading process length (Fig. 1E). Overall, these data suggest that the extension of the leading process increases the plasma membrane tension.

## Increased plasma membrane tension triggers $Ca^{2+}$ signaling through mechanosensitive ion channels

It is well-established that mechanosensitive ion channels trigger cell signaling in response to extracellular mechanical stimuli (Lee et al, 1999; Douguet and Honore, 2019). We hypothesized that leading process extension may serve as an endogenous mechanical stimulus to trigger cell signaling. To test this possibility, we examined the relationship between leading process extension and $Ca^{2+}$ signaling in 3D Matrigel. The intracellular $Ca^{2+}$ concentration ($[Ca^{2+}]_i$) of olfactory interneurons was monitored using the $Ca^{2+}$ indicator CalRedR525/650 (Fig. 2A; Movie EV1). The average length of the

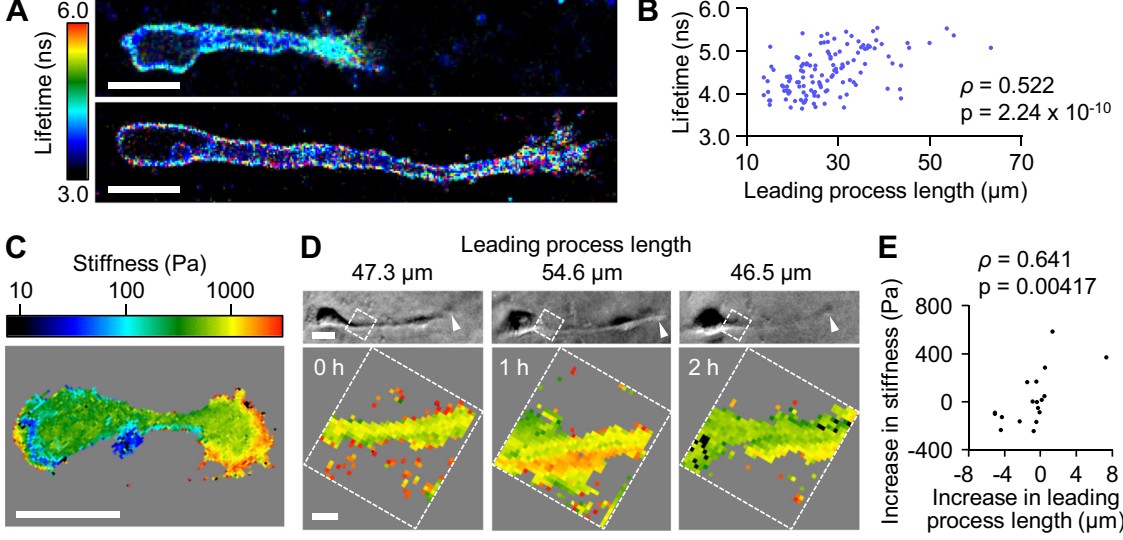

**Figure 1. Leading process extension increases plasma membrane tension.**

(A) Fluorescence lifetime images of Flipper-TR in olfactory interneurons migrating in 3D Matrigel with leading processes of different lengths. The color bar indicates the lifetime in nanoseconds (ns). (B) Fluorescence lifetime of Flipper-TR in the leading process shaft and soma plotted against the leading process length. $n = 129$ cells. Two-tailed Spearman's correlation coefficient ($\rho$) was calculated to determine the relationship between the membrane tension and the leading process length. (C) A stiffness map of an olfactory interneuron on 2D laminin-coated substrate produced by SICM. (D) Bright field images (upper) and stiffness maps (lower) of a migrating olfactory interneuron on 2D laminin-coated substrate acquired at 1-h intervals. The arrowheads indicate the growth cone. The stiffness of the plasma membrane was measured at the proximal leading process indicated by the boxes. (E) The increase in the stiffness plotted against the increase in the leading process length. The first measurements were set as 0 as controls and omitted from the plot. $n = 27$ measurements from 9 cells. Two-tailed Spearman's correlation coefficient ($\rho$) was calculated to determine the relationship between the stiffness and the leading process length. Scale bars, 10 μm (A, C and D, upper); 2 μm (D, lower). See also Appendix Fig. S1. Source data are available online for this figure.

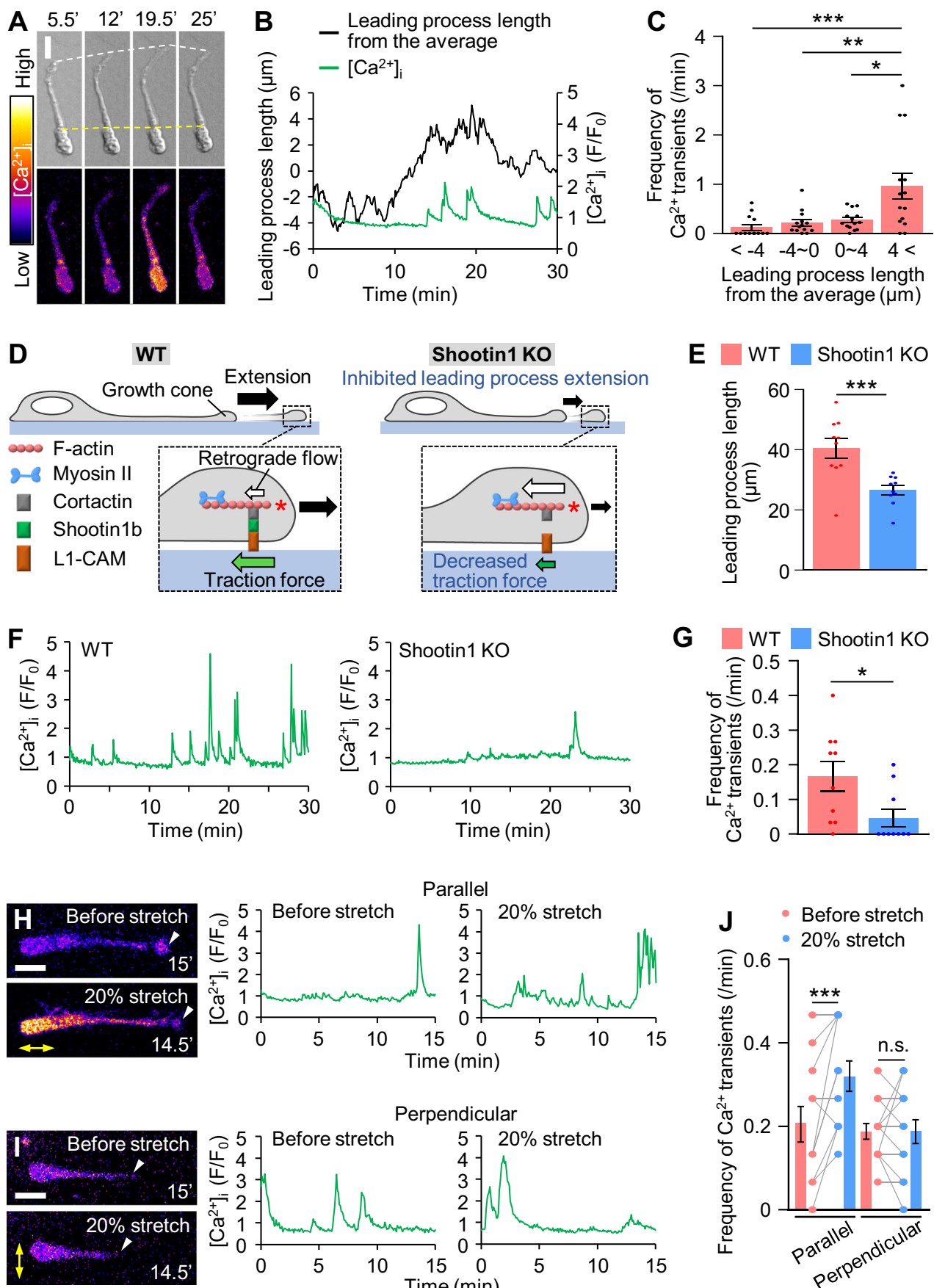

**Figure 2. Leading process extension triggers Ca²⁺ transients.**

(A) Time-lapse DIC (upper) and $[Ca^{2+}]_i$ (lower) images of a migrating olfactory interneuron loaded with 1 µM CalRed R525/650. $[Ca^{2+}]_i$ is indicated as the ratio of the fluorescence intensity at 525 nm to that at 650 nm. The white and yellow dashed lines indicate the fronts of the growth cone and soma, respectively. See Movie EV1. (B) Time course of the leading process length and $[Ca^{2+}]_i$ of the neuron shown in (A). The average leading process length was set as 0. See Appendix Fig. S2A. (C) Relationship between the average leading process length and the frequency of Ca²⁺ transients. The frequency of Ca²⁺ transients was measured during increases and decreases in the neurite length. $n = 56$ measurements from 14 cells. $p = 0.00088$ (< -4 vs 4<); $p = 0.0101$ (-4-0 vs 4<); $p = 0.0382$ (0-4 vs 4<). (D) The mechanism of shootin1b-mediated leading process extension (Minegishi et al, 2018). At the leading process growth cone, actin filaments (F-actins) undergo retrograde flow (white arrow) powered by actin polymerization (asterisk) and myosin II contraction. Shootin1b couples F-actin retrograde flow and the extracellular substrate through its interactions with an actin binding protein cortactin and a cell adhesion molecule L1-CAM. The movement of F-actin flow is transmitted through shootin1b to the substrate, thereby generating traction force on the substrate (green arrow). Traction force drives leading process extension (black arrows, left). Disruption of actin-substrate coupling by shootin1 KO decreases traction force, and inhibits leading process extension (right). (E) Leading process length of migrating WT ($n = 10$ cells) and shootin1 KO ($n = 10$ cells) olfactory interneurons. $p = 0.00237$. (F) Time courses of $[Ca^{2+}]_i$ in migrating WT (left) and shootin1 KO (right) olfactory interneurons. Neurons were imaged at 5-sec intervals for 30 min. See Appendix Fig. S2B. (G) The frequency of Ca²⁺ transients in migrating WT ($n = 10$ cells) and shootin1 KO ($n = 10$ cells) neurons in (F). $p = 0.0241$. (H, I) Representative images of $[Ca^{2+}]_i$ of migrating olfactory interneurons (left) and the time courses of $[Ca^{2+}]_i$ (right). The arrowheads indicate growth cones. Neurons cultured on elastic chambers were imaged at 5-sec intervals for 15 min before (control) and after 20% stretch (yellow arrows). The arrowheads indicate the growth cones. We selected migrating olfactory interneurons with leading processes oriented parallel (H) or perpendicular (I) to the stretch direction. See Appendix Fig. S4 and Movie EV2. (J) The frequency of Ca²⁺ transients in migrating neurons before and after 20% stretch. Parallel, $n = 11$ cells; perpendicular, $n = 12$ cells. $p = 0.00805$ (parallel); $p > 0.999$ (perpendicular). Data represent mean ± SEM. Statistical analysis was performed using the two-tailed Mann–Whitney $U$-test (C), two-tailed unpaired Welch's $t$-test (E), two-tailed unpaired Student's $t$-test (G), two-tailed paired $t$-test (J). *$p < 0.05$; **$p < 0.02$; ***$p < 0.01$; n.s., no significant difference. Scale bars, 10 µm (A, H and I). See also Appendix Figs. S2–S4. Source data are available online for this figure.

leading process was $39.0 \pm 2.7$ µm (mean ± SEM, $n = 14$ cells), while the length of neurite extension and retraction was $11.3 \pm 0.7$ µm ($29.0 \pm 1.8\%$ of the neurite length). Remarkably, the frequency of the Ca²⁺ transients increased when the leading process length was 4 µm larger than the average length (Fig. 2B,C; Appendix Fig. S2A).

The extension of the leading process is propelled by the traction force produced by the growth cone at its tip (Jiang et al, 2015; Minegishi et al, 2018; Umeshima et al, 2019; Minegishi and Inagaki, 2020). We previously reported that shootin1b mediates the production of the traction force at the leading process growth cone (Minegishi et al, 2018) (Fig. 2D). Importantly, the reduction in the process length by shootin1 knockout (KO) decreased the frequency of Ca²⁺ transients (Fig. 2E-G; Appendix Fig. S2B). We also increased the leading process length using a cell-stretching device (Appendix Fig. S3) (Tsujita et al, 2021); increasing process length by 20% increased the frequency of Ca²⁺ transients (Fig. 2H,J; Appendix Fig. S4A; Movie EV2). On the other hand, no effect was observed when the stretch was applied perpendicular to the leading process (Fig. 2I,J; Appendix Fig. S4B); we expect that perpendicular stretch cannot efficiently stretch the leading process shaft. Furthermore, treatment of neurons with 5 µM GsMTx4, which inhibits various mechanosensitive ion channels, including Piezo and TRP channels (Gnanasambandam et al, 2017; Suchyna, 2017), reduced the frequency of Ca²⁺ transients (Fig. 3A; Appendix Fig. S5). Thus, we conclude that the increased plasma membrane tension caused by the leading process extension triggers Ca²⁺ signaling through mechanosensitive ion channels.

## Tmem63b mediates tension-triggered Ca²⁺ signaling

To identify the molecule involved in the tension-mediated Ca²⁺ signaling process, we analyzed the expression of mechanosensitive ion channels through quantitative PCR (qPCR). We found that several mechanosensitive ion channels, including Tmem63b (Murthy et al, 2018), are expressed in olfactory interneurons (Fig. 3B). Although there are currently no reports showing that GsMTx4 inhibits Tmem63b, we focused on this ion channel because of its high expression. We performed immunohistochemistry using an anti-Tmem63b antibody (Fig. 3C), and detected

Tmem63b expression in the migratory pathway of olfactory interneurons, including the subventricular zone (SVZ, asterisk), rostral migratory stream (RMS, arrowheads) and olfactory bulb. In addition, Tmem63b was widely detected along the leading process of doublecortin-positive olfactory interneurons (Fig. 3D). Tmem63 was localized at the plasma membrane (arrowheads) as well as in the cytoplasm, presumably in the lysosome (Li et al, 2024). Repression of Tmem63b by microRNA (Appendix Fig. S6A) suppressed spontaneous Ca²⁺ transients (Fig. 3E,F,H, red asterisks) and abolished tension-triggered Ca²⁺ transients (Fig. 3H, blue n.s.). Furthermore, expression of RNAi-refractory Tmem63b rescued the inhibition of spontaneous and tension-triggered Ca²⁺ transients (Fig. 3G,H), indicating that Tmem63b is involved in tension-triggered Ca²⁺ signaling by olfactory interneurons. The activation of Tmem63b, which is activated by an increase in plasma membrane tension (Murthy et al, 2018), further supports our conclusion that the extension of the leading process increases the plasma membrane tension.

## Tension-triggered Ca²⁺ signaling drives somal translocation through myosin II activation

Ca²⁺ signaling activates the myosin light chain kinase-myosin II pathway (Horigane et al, 2019). Therefore, we examined whether the activation of Tmem63b triggers actomyosin contraction and somal translocation. Immunostaining analyses of phosphorylated myosin light chain (pMLC) showed that stretching of the leading process elevated myosin II activity at the rear of the cell (Fig. 4A,B; Appendix Fig. S6B). Furthermore, this activation was abolished by Tmem63b knockdown (Fig. 4A,B), indicating that tension-triggered Ca²⁺ signaling activates myosin II. When Tmem63b was knocked down, neurons extended longer leading processes (Fig. 5A–C; Appendix Fig. S7). However, despite the significant process extension, the soma did not translocate efficiently in 3D Matrigel (Fig. 5A; Movie EV3). Neurons showed slower somal translocation speeds than control cells (Fig. 5A,B,D; Appendix Fig. S7), which was accompanied by a significant reduction in the neuronal migration speed (Fig. 5A,E). Similar inhibition was observed in neurons treated with the myosin II inhibitor

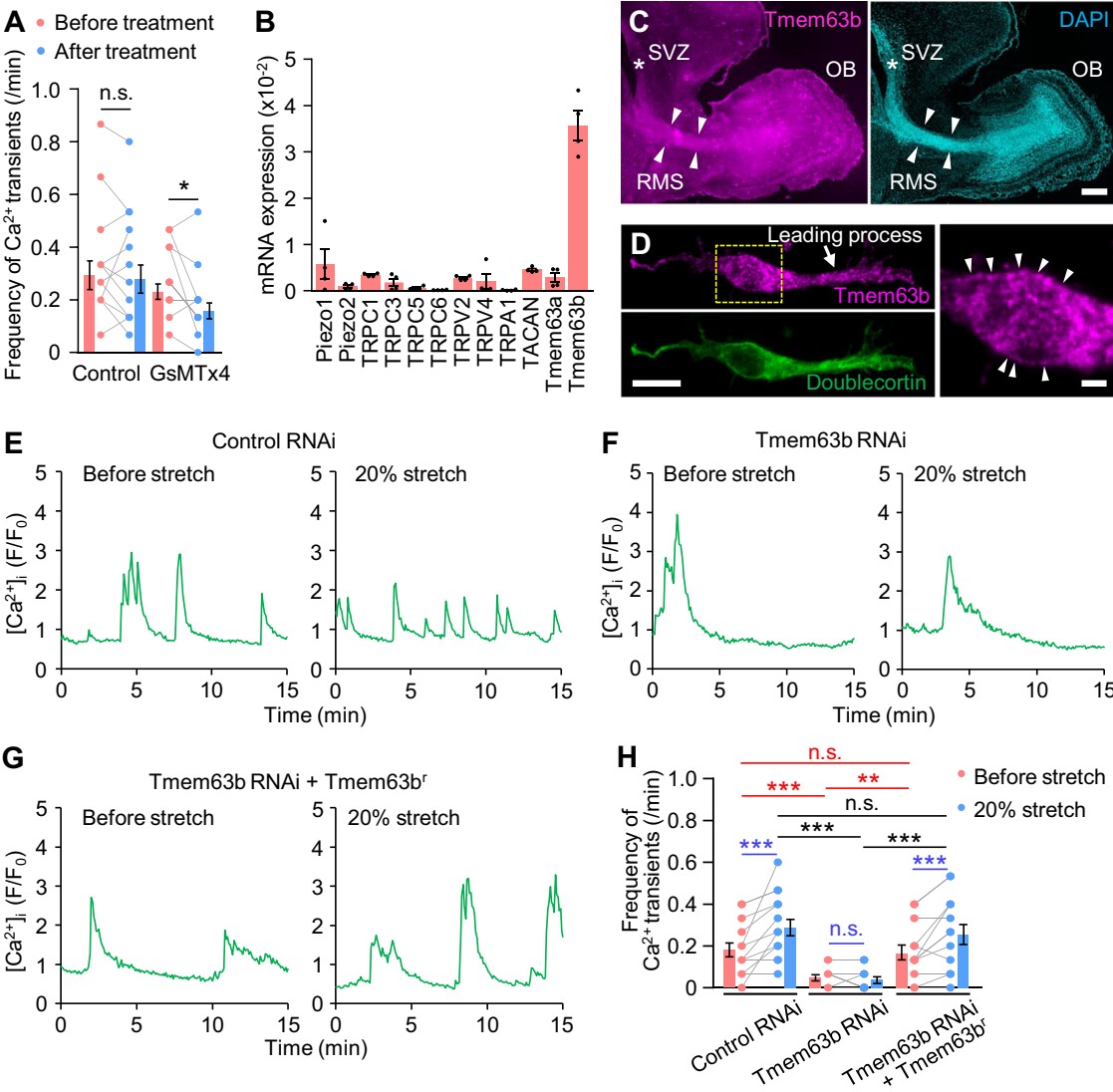

**Figure 3. Tmem63b mediates stretch-induced Ca²⁺ transients.**

(A) The frequency of Ca²⁺ transients in migrating olfactory interneurons before and after treatment with vehicle control ($n = 12$ cells) or 5 μM GsMTx4 ($n = 13$ cells). $p = 0.322$ (control); $p = 0.0234$ (GsMTx4). See Appendix Fig. S5. (B) Expression of mechanosensitive ion channels in olfactory interneurons. The mRNA of the mechanosensitive ion channels was quantified with qPCR and normalized by the expression level of GAPDH. $n = 4$ independent experiments. See Table EV1. (C) A sagittal section of postnatal day 5 mouse brain labeled with anti-Tmem63b antibody and DAPI. OB, olfactory bulb. RMS, rostral migratory stream. SVZ, subventricular zone. An asterisk and arrowheads indicate SVZ and RMS, respectively. Scale bar, 200 μm. (D) An olfactory interneuron immunostained by anti-Tmem63b and anti-doublecortin antibodies. Anti-doublecortin antibody was used as a marker of olfactory interneurons. Arrowheads indicate Tmem63b localization at the plasma membrane. Scale bars, 10 μm (left); 2 μm (right). (E–G) Time courses of $[Ca^{2+}]_i$ in migrating olfactory interneurons expressing control microRNA (E), Tmem63b microRNA (F) or Tmem63b microRNA + RNAi-refractory Tmem63b (Tmem63b^r) (G). Migrating olfactory interneurons cultured on elastic chambers were imaged at 5-s intervals for 15 min before (control) and after uniaxial stretch (20% stretch). We imaged olfactory interneurons with leading processes oriented parallel to the stretch direction. (H) Tmem63b knockdown abolishes the extension-triggered Ca²⁺ transient. The frequency of Ca²⁺ transients in migrating olfactory interneurons before and after 20% stretch. Control microRNA, $n = 13$ cells; Tmem63b microRNA, $n = 12$ cells; Tmem63b microRNA + RNAi-refractory Tmem63b (Tmem63b^r), $n = 12$ cells. $p = 0.0045$ (control microRNA, before stretch vs 20% stretch); $p = 0.339$ (Tmem63b microRNA, before stretch vs 20% stretch); $p = 0.00819$ (Tmem63b microRNA + Tmem63b^r, before stretch vs 20% stretch); $p = 0.00768$ (control microRNA vs Tmem63b microRNA, before stretch); $p = 0.0174$ (Tmem63b microRNA vs Tmem63b microRNA + Tmem63b^r, before stretch); $p = 0.962$ (control microRNA vs Tmem63b microRNA + Tmem63b^r, before stretch); $p = 0.000109$ (control microRNA vs Tmem63b microRNA, 20% stretch); $p = 0.000859$ (Tmem63b microRNA vs Tmem63b microRNA + Tmem63b^r, 20% stretch); $p = 0.804$ (control microRNA vs Tmem63b microRNA + Tmem63b^r, 20% stretch). Data represent mean ± SEM. Statistical analysis was performed using the two-tailed Wilcoxon signed-rank test (A), two-tailed paired t-test (H, blue), and two-tailed one-way ANOVA with Tukey's post hoc test (H, red and black). $*p < 0.05$; $**p < 0.02$; $***p < 0.01$; n.s., no significant difference. Source data are available online for this figure.

blebbistatin (100 μM) or MLC kinase inhibitor ML-7 (10 μM) (Appendix Fig. S8). Furthermore, expression of RNAi-refractory Tmem63b in neurons expressing Tmem63b microRNA rescued extension-triggered myosin II activation (Fig. 4A,B) as well as the somal translocation and migration speed in 3D Matrigel (Fig. 5; Appendix Fig. S7; Movie EV3). Taken together, these data indicate that tension-triggered Ca²⁺ signaling drives somal translocation through myosin II activation.

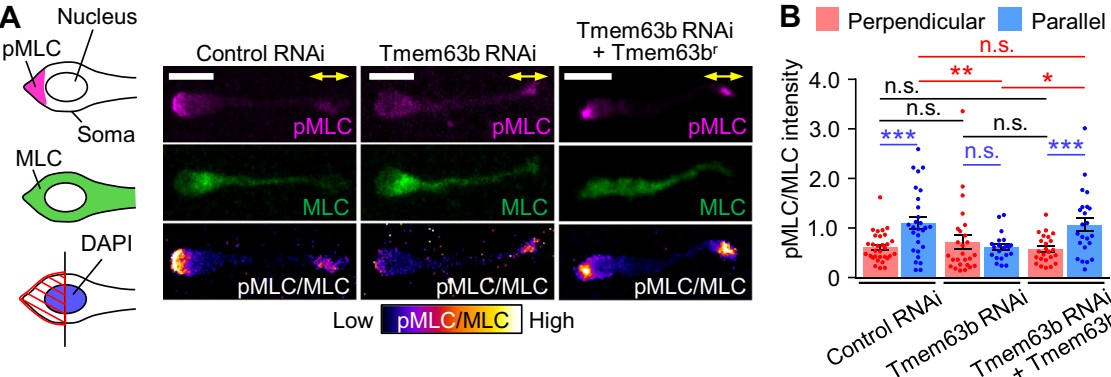

**Figure 4. Tmem63b mediates stretch-induced myosin II activation.**

(A) Immunocytochemical analysis of myosin II activity in olfactory interneurons expressing control microRNA, Tmem63b microRNA, or Tmem63b microRNA + Tmem63b[r]. Olfactory interneurons cultured on elastic chambers were fixed after 20% stretch, and then stained with anti-phospho-myosin light chain 2 (pMLC) antibody (magenta), anti-myosin light chain 2 (MLC) antibody (green) and DAPI. Neurons with leading processes oriented parallel to the stretch direction (yellow arrows) were analyzed. A diagram describing the analysis of myosin II activity in olfactory interneurons (left). Fluorescence images of pMLC, MLC, and DAPI in olfactory interneurons were acquired. We then drew a line that cross the middle of nucleus stained by DAPI, and measured the fluorescence intensities of pMLC and MLC within the posterior half of the soma (shaded area). The myosin II activity was calculated as the ratio of the fluorescence intensity of pMLC to that of MLC. The ratio of the fluorescence intensity of pMLC to that of MLC (pMLC/MLC) is displayed by the pseudocolor bar. Scale bars, 10 μm. (B) Statistical analysis of myosin II activity in olfactory interneurons from the data shown in (A). Myosin II activity (pMLC/MLC) was analyzed at the rear of the soma. Control microRNA: perpendicular, $n = 31$ cells; parallel, $n = 28$ cells. Tmem63b microRNA: perpendicular, $n = 26$ cells; parallel, $n = 21$ cells. Tmem63b microRNA + Tmem63b[r]: perpendicular, $n = 24$ cells; parallel, $n = 23$ cells. $p = 0.00105$ (control microRNA, perpendicular vs parallel); $p = 0.464$ (Tmem63b microRNA, perpendicular vs parallel); $p = 0.00474$ (Tmem63b microRNA + Tmem63b[r], perpendicular vs parallel); $p > 0.999$ (control microRNA vs Tmem63b microRNA, perpendicular); $p > 0.999$ (Tmem63b microRNA vs Tmem63b microRNA + Tmem63b[r], perpendicular); $p > 0.999$ (control microRNA vs Tmem63b microRNA + Tmem63b[r], perpendicular); $p = 0.0147$ (control microRNA vs Tmem63b microRNA, parallel); $p = 0.031$ (Tmem63b microRNA vs Tmem63b microRNA + Tmem63b[r], parallel); $p > 0.999$ (control microRNA vs Tmem63b microRNA + Tmem63b[r], parallel). Data represent mean ± SEM. Statistical analysis was performed using the two-tailed Mann–Whitney U-test (B, blue), and two-tailed Kruskal–Wallis test with Dunn's post hoc test (B, red and black). $*p < 0.05$; $**p < 0.02$; $***p < 0.01$; n.s., no significant difference. See also Appendix Fig. S6. Source data are available online for this figure.

# Discussion

In this study, we have shown that leading process extension of migrating olfactory interneurons increases plasma membrane tension. This tension elevation activates the mechanosensitive ion channel Tmem63b and triggers $Ca^{2+}$ signaling, which drives somal translocation through actomyosin contraction.

## Leading process extension increases plasma membrane tension for $Ca^{2+}$ signaling

Our data showed that the stretch along the leading process shaft, not perpendicular stretching, triggered $Ca^{2+}$ influx through Tmem63b (Fig. 2H–J). The perpendicular stretching stretches the growth cone and the soma. On the other hand, the parallel stretching stretches the leading process in addition to the growth cone and soma. Thus, our data indicate that the stretch along the leading process shaft, rather than that at the growth cone and the soma, plays the main role in Tmem63b activation. Indeed, changes in leading process length are positively correlated with changes in plasma membrane tension along the leading process (Fig. 1) and Tmem63b is widely localized along the leading process (Fig. 3D). In addition, previous studies reported that gradients of the plasma membrane tension is formed between the axonal growth cone and the soma (Popov et al, 1993; Dal and Sheetz, 1995) and that the tension propagates rapidly along axons (Gomis Perez et al, 2022; Shi et al, 2022). Consistently, tension-provoked $Ca^{2+}$ transients were widely detected in the neurons (Fig. 2H), as reported (Martini and Valdeolmillos, 2010).

In vivo, the present tension-induced $Ca^{2+}$ influx may be affected by the extracellular environment. Mechanosensitive channels trigger cell signaling in response to mechanical stimuli from the extracellular environment, including cell compression, shear stress, substrate deformation, and osmotic changes (Lee et al, 1999; Murthy et al, 2018; Douguet and Honore, 2019; Du et al, 2020). These stimuli along the migration pathway may modulate the movement of olfactory interneurons. The analysis of the membrane tension of migrating neurons in vivo remains an important issue for future studies.

## Tension-induced signaling for efficient 3D somal translocation

In keratocytes and neutrophils, it has been proposed that plasma membrane tension itself assists translocation of the cell rear forward, thereby propelling cell migration (Ofer et al, 2011; Tsai et al, 2019). However, our data in 3D environments have shown that blockade of the tension-mediated actomyosin activation by Tmem63b knockdown, as well as by the inhibition of MLCK or myosin II, disturbed somal translocation and inhibited neuronal migration in 3D Matrigel (Fig. 5; Appendix Fig. S8). These data suggest that plasma membrane tension is not sufficient to translocate the neuronal soma against the mechanical barriers of the surrounding 3D environment. In order to drive efficient somal translocation in the 3D environment, neurons must generate robust forces through the tension-mediated signaling pathway.

Although the leading process extension provokes $Ca^{2+}$ transient widely in migrating neurons, the downstream effector actomyosin

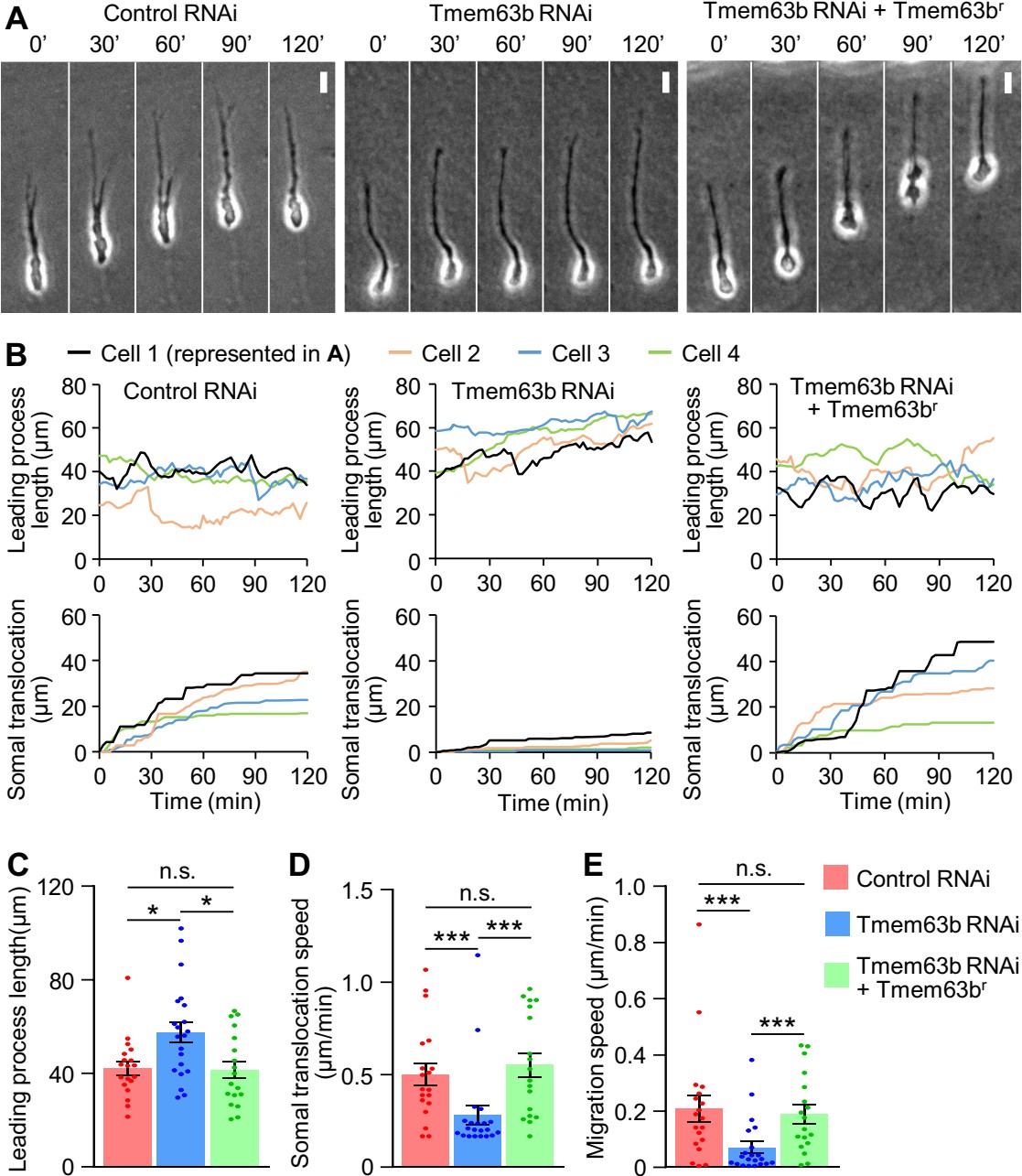

**Figure 5. Tmem63b knockdown inhibits saltatory and efficient neuronal migration.**

(A) Time-lapse images of migrating olfactory interneurons expressing control microRNA, Tmem63b microRNA or Tmem63b microRNA + Tmem63b$^r$. Images were acquired at 2-min intervals for 120 min. See Movie EV3. Scale bar, 10 μm. (B) Leading process length (upper) and somal translocation (lower) of olfactory interneurons. Black lines indicate the cell represented in (A). (C–E) Leading process length (C), somal translocation speed (D) and migration speed (E) of olfactory interneurons shown in (A). Control microRNA, $n = 19$ cells; Tmem63b microRNA, $n = 21$ cells; Tmem63b microRNA + Tmem63b$^r$, $n = 18$ cells. (C) $p = 0.0411$ (control microRNA vs Tmem63b microRNA); $p = 0.0239$ (Tmem63b microRNA vs Tmem63b microRNA + Tmem63b$^r$); $p > 0.999$ (control microRNA vs Tmem63b microRNA + Tmem63b$^r$). (D) $p = 0.00279$ (control microRNA vs Tmem63b microRNA); $p = 0.000711$ (Tmem63b microRNA vs Tmem63b microRNA + Tmem63b$^r$); $p > 0.999$ (control microRNA vs Tmem63b microRNA + Tmem63b$^r$). (E) $p = 0.00985$ (control microRNA vs Tmem63b microRNA); $p = 0.00863$ (Tmem63b microRNA vs Tmem63b microRNA + Tmem63b$^r$); $p > 0.999$ (control microRNA vs Tmem63b microRNA + Tmem63b$^r$). Data represent mean ± SEM. Statistical analysis was performed using the two-tailed Kruskal–Wallis test with Dunn's post hoc test (C–E). *$p < 0.05$; ***$p < 0.01$; n.s., no significant difference. See also Appendix Figs. S7 and S8. Source data are available online for this figure.

is localized at the rear of the soma (Martini and Valdeolmillos, 2010). Thus, the global activation of Tmem63b triggers a local actomyosin contraction at the rear. In addition, migrating neurons form a cage-like microtubule structure around the nucleus, which act as the scaffold for dynein-mediated nucleokinesis (Rivas and Hatten, 1995; Schaar and McConnell, 2005; Tsai et al, 2007; Vallee et al, 2009). The effects of Ca$^{2+}$ transients on the growth and stability of perinuclear microtubules in migrating neurons remains

unclear. Similarly, the coordination between the actin and microtubule dynamics around the nucleus is yet to be elucidated. Analyzing the potential link between the tension-induced mechanical signaling and perinuclear microtubule dynamics is an interesting subject for future work.

## Integrated model of neuronal migration

By providing a missing link to supplement previous findings (Minegishi and Inagaki, 2020), our work presents an integrated model of neuronal migration (Fig. 6) involving a series of biochemical (magenta) and mechanical (blue) steps (①–⑨). (i) Shootin1b accumulation at the growth cone generates the traction force and triggers leading process extension (Minegishi et al, 2018). (ii) The leading process extension increases the plasma membrane tension, which activates Tmem63b and triggers $Ca^{2+}$ signaling. (iii) Upon activation by $Ca^{2+}$ signaling, actomyosin produces contractile forces, pushing the nucleus forward and driving somal translocation (Komuro and Rakic, 1996; Martini and Valdeolmillos, 2010). This is followed by shootin1b accumulation at the growth cone, initiating the next cycle. Olfactory interneurons express several mechanosensitive channels, including Piezo and TRPC channels

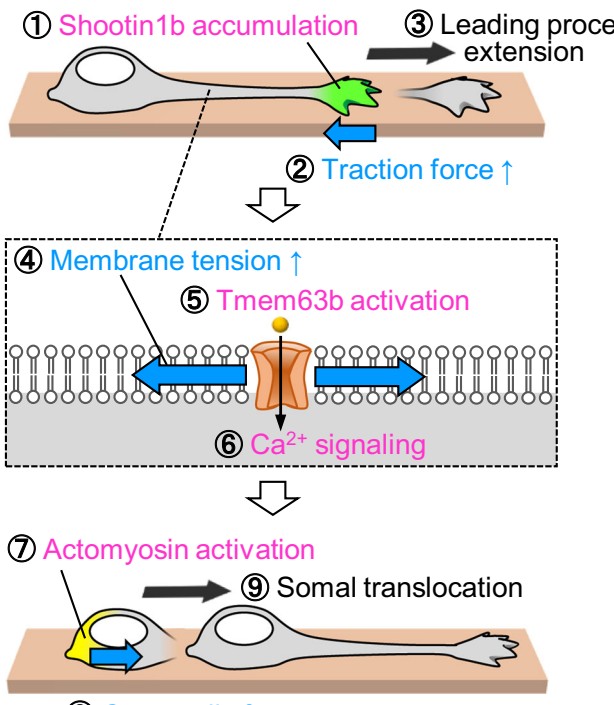

**Figure 6. An integrated model of saltatory neuronal migration, involving a series of biochemical and mechanical steps (①–⑨).**

Shootin1b accumulates at the leading process growth cone (green) and couples F-actin retrograde flow and the extracellular substrate, thereby generating traction force to drive leading process extension (see Fig. 2D). The increase in the process length elevates the plasma membrane tension. The elevated membrane tension activates Tmem63b, which induces $Ca^{2+}$ influx and triggers $Ca^{2+}$ signaling. Upon activation induced by $Ca^{2+}$ signaling, actomyosin (yellow) produces contractile forces, pushing the nucleus forward and driving somal translocation.

(Fig. 3B), and their inhibitor GsMTx4 (Gnanasambandam et al, 2017; Suchyna, 2017) partially inhibits $Ca^{2+}$ transients in these cells (Fig. 3A). We speculate that other mechanosensitive channels, including Piezo and TRPC channels, may also mediate the tension-induced mechanical signaling for somal translocation.

Neurons migrate by repeating leading process extension and somal translocation. $Ca^{2+}$ transients are required for somal translocation not only of olfactory interneurons but also of cortical neurons, cerebellar granule cells, and gonadotropin-releasing hormone-expressing neurons (Komuro and Rakic, 1996; Martini and Valdeolmillos, 2010; Hutchins et al, 2013; Rash et al, 2016). Furthermore, somal translocation of these neurons is driven by actomyosin and dynein motor complexes (Schaar and McConnell, 2005; Metin et al, 2006; Tsai et al, 2007; Solecki et al, 2009; Vallee et al, 2009; Martini and Valdeolmillos, 2010; Hutchins et al, 2013; Jheng et al, 2018). Therefore, the present mechanical signaling could underlie the cell migration of a wide range of neuronal types.

## Possible roles of Tmem63b in brain development and neural network formations

Defects in neuronal migration cause brain diseases, including brain malformations, epilepsy, and intellectual disability (Ross and Walsh, 2001; Valiente and Marin, 2010; Evsyukova et al, 2013; Stouffer et al, 2016; Nakajima et al, 2021). Recently, heterozygous mutations of *TMEM63b* have been identified in patients with severe early-onset developmental and epileptic encephalopathies and progressive neurodegeneration (Vetro et al, 2023). These patients exhibit brain malformations, including multiregional or widespread white matter, dysmorphic lateral ventricles, thinning of the corpus callosum, cerebellar atrophy, and atrophy of the cerebral cortex (Vetro et al, 2023). Cells expressing the mutant Tmem63b have a reduced response to membrane stretch (Vetro et al, 2023), suggesting a possible linkage between the neuronal migration defects caused by Tmem63b dysregulation and brain malformations.

After migration, neurons undergo dynamic morphological changes to perform their activities, including, polarity formation, axon guidance, synapse formation, synaptic plasticity, and neural network regeneration. Similar mechanical signaling, through membrane tension and mechanosensitive channels, could occur during these processes as they can also alter the plasma membrane tension locally or globally.

## Methods

### Animals

All relevant aspects of the experimental procedures were approved by the Institutional Animal Care and Use Committee of Nara Institute of Science and Technology. Postnatal day 5 (P5) C57BL/6 mice were obtained from Japan SLC and CLEA Japan. P5 Shootin1 KO pups were obtained by crossing male and female shootin1 heterozygous mice; the offspring genotypes were checked by PCR with the following primers: Genotyping F1 (5′-CAGACTGCTACC-CACTACCCCCTAC-3′) and Genotyping R1 (5′-CCTAGAGCTG-GACAGCGGATCTGAG-3′) for the WT allele; Genotyping F2 (5′-CCCAGAAAGCGAAGGAACAAAGCTG-3′) and Genotyping R2

(5′-ACCTTGCTCCTTCAAGCTGGTGATG-3′) for the shootin1 KO allele. Shootin1 KO pups and their littermates were bred with their mother under standard conditions (12 h/12 h light/dark cycle, access to dry food and water). Mice of both sexes were used for experiments. The generation of shootin1 KO mice is described elsewhere (Baba et al, 2018). Chimeric mice were crossed with C57BL/6 mice for at least nine generations before analysis.

## Culture of dissociated SVZ-derived olfactory interneurons and transfection

SVZ tissues dissected from P5 mouse brains were dissociated with papain (Nacalai Tesque) as previously described (Minegishi et al, 2018). The dissociated olfactory interneurons were reaggregated in neurobasal medium (Thermo Fisher Scientific) containing 10% fetal bovine serum (Thermo Fisher Scientific), 2% B-27 supplement (Thermo Fisher Scientific), 1 mM glutamine (Nacalai Tesque) and 100 U/mL penicillin–streptomycin (Nacalai Tesque) for 5–7 h. The aggregates were then embedded in a mixture of 75% Matrigel (Corning) and 25% Leibovitz′s L-15 (L-15) medium (Thermo Fisher Scientific), plated on glass bottom dishes (Matsunami) or elastic silicon chambers (Strex, catalog number STB-CH-04), and cultured in neurobasal medium containing 2% B-27 supplement, 1 mM glutamine and 100 U/mL penicillin–streptomycin in a humidified 5% $CO_2$ incubator at 37 °C. For cell stiffness measurements by SICM, the dissociated olfactory interneurons were cultured on glass bottom dishes coated subsequently with 100 μg/mL poly-D-lysine (Sigma) and 5 μg/mL laminin (Wako Pure Chemical Industries) as previously described (Minegishi et al, 2018). For RNAi experiments, olfactory interneurons were transfected with vectors using Nucleofector (Lonza) before reaggregation. For the immunoblot analysis shown in Appendix Fig. S6A, transfected olfactory interneurons were cultured on poly-D-lysine-coated plastic dishes.

## DNA construction

To generate a Tmem63b microRNA expressing vector, we used a Block-iT Pol II miR RNAi expression vector kit (Thermo Fisher Scientific). The targeting sequence of Tmem63b (5′-CGAGGACAGCCTCATAGAGAA-3′, corresponding to nucleotides 2424–2444 in the coding region of mouse Tmem63b) was cloned and inserted into the pcDNA6.2-GW/EmGFP-miR expression vector. To prevent signal overlap between the GFP reporter of the microRNA expression vector and the fluorescent dyes used in our experiments, a DNA cassette containing the microRNA sequence was subcloned and inserted into pEBFP-C1 (Takara Bio) and pmCherry-C1 (Takara Bio) for $Ca^{2+}$ imaging and immunocytochemical analyses, respectively. Full-length cDNA of mouse Tmem63b was obtained by PCR amplification of mouse subventricular zone (SVZ) cDNA with the primers Tmem63b Fw (5′-GACGACGATGACAAGAG ATCTATGCTGCCGTTCTTGCTGGCCACA-3′) and Tmem63b Rv (5′-GATACGCGTGGTACCAGATCTTTACTGGTGAATCT-CATTCTCTAT-3′), and subcloned and inserted into pCMV5-FLAG vector (Agilent Technology). The mouse SVZ cDNA was prepared as described in the 'qPCR analysis of migrating neurons' section. The RNAi refractory Tmem63b mutant was generated by inducing seven silent mutations in nucleotides 2424-2444 of Tmem63b (5′-T̲GA̲A̲GA-TA̲GT̲CT̲GA̲T̲TGA̲AAA-3′; the underlines indicate the mutations) by PCR.

## Analysis of plasma membrane tension by Flipper-TR

After 2 days of culture in Matrigel, olfactory interneurons were incubated with 1 μM of the membrane tension-sensitive probe Flipper-TR (Spirochrome) diluted in culture medium for 1 h at 37 °C and 5% $CO_2$. Then the culture medium was replaced with L-15 medium containing 2% B-27 supplement, 1 mM glutamine, and 100 U/mL penicillin–streptomycin. Fluorescence lifetime imaging microscopy (FLIM) analysis was performed at 37 °C using a confocal laser scanning microscope (SP8, Leica) equipped with a Fast Lifetime Contrast (FALCON) module, HC PL APO CS2 100x/1.40 oil immersion lens, and imaging software Las-X (Leica). The cells were excited with a pulsed 488 nm laser operating at 80 MHz, and the emission signals were detected at 575–625 nm. For quantification, we defined the cell regions and measured the average lifetime within this region as described in Appendix Fig. S1. A double exponential fit was performed to extract lifetimes: as previously described (Colom et al, 2018), the longer lifetime was used for the plots in Fig. 1B. We did not perform any cutoff for the fits.

## Time-lapse imaging of plasma membrane tension by SICM

As an increase in the plasma membrane tension leads to decreased membrane deformability (Diz-Munoz et al, 2013), membrane stiffness has been used as an indicator of plasma membrane tension. SICM can be used to measure membrane stiffness without directly contacting the membrane and has been used to estimate membrane tension (Rheinlaender and Schaffer, 2013; Bednarska et al, 2020). In SICM, a glass nanopipette is used as a probe to monitor ion currents between an Ag/AgCl electrode placed inside the pipette and an Ag/AgCl electrode located in a bath. A constant pressure is applied to the pipette to generate a microfluidic flow, which deforms the cell surface. As the pipette approaches the cell, the ion current decreases. This decrease is slower for softer cells because the microfluidic flow progressively indents the cell surface. Therefore, the plasma membrane stiffness can be determined based on the ion current approach curve (Rheinlaender and Schaffer, 2013).

The plasma membrane tension was measured using a home-made SICM system with an inverted optical microscope (Eclipse-Ti, Nikon). The setup of the SICM system, the scanning algorithms and the protocol were described previously (Takahashi et al, 2020). Before the measurements were acquired, the culture medium was replaced with L-15 medium containing 2% B-27 supplement, 1 mM glutamine and 100 U/mL penicillin–streptomycin. Glass nanopipettes (aperture inner radius 100 nm) filled with L-15 medium were used as probes. Scanning was performed in hopping mode with the following parameters: hopping amplitude, 3–5 μm; waiting time after lateral movement, 1 ms (during this time, a reference current was measured as the average of the direct current through the probe); probe approach and withdrawal speed, 20 and 500–1000 nm/ms, respectively; applied pressure, 10 kPa; and set point, 98% of the reference current. We recorded ion currents and the vertical probe position as the probe approach the cell surface. For the experiments shown in Fig. 1C, we obtained data for a 25 × 25 μm² area containing 128 × 128 points. For the experiments shown in Fig. 1D,E, we obtained data for a 10 × 10 μm² area containing 32 × 32 points, and repeated the experiment 3 times for

each cell at 1 h intervals. After the measurements were performed, optical cell images were acquired using a CSI S Plan Fluor ELWD 40x/0.6 NA objective lens (Nikon), a complementary metal oxide semiconductor (CMOS) camera (ORCA Flash 4.0 C13440-20CU, Hamamatsu) and imaging software (HCI image, Hamamatsu) or using a Plan Flour 10x/0.30 NA objective lens (Nikon), a CMOS camera (Wraycam-noa2000, Wraymer) and imaging software (MicroStudio, Wraymer). The SICM data were analyzed and stiffness maps were produced using a handmade program developed based on LabVIEW2014 (National Instruments). The approach curve of each point was plotted as the ion current versus the vertical probe position, and the slope between 98.5% and 99% of the reference current was determined with a line fit. Young's modulus of the cell surface was determined as described previously (Rheinlaender and Schaffer, 2013):

$$E = p_0 A \left( \frac{s_\infty}{s} - 1 \right)^{-1}$$

where $E$ is Young's modulus, $p_0$ is the applied pressure, $A$ is the pipette geometrical parameter (whose value is 0.4, as determined previously (Rheinlaender and Schaffer, 2013)), and $s$ and $s_\infty$ are the measured slopes on the cell surface and substrate, respectively.

## Ca²⁺ imaging

$[Ca^{2+}]_i$ was monitored using the ratiometric calcium indicator dye CalRed R525/650 AM (AAT Bioquest); when this indicator is excited at 488 nm upon $Ca^{2+}$ binding, the emission signals increased at 525 nm and decreased at 650 nm. After 2 days in culture, olfactory interneurons were loaded with 1 μM CalRed R525/650 AM, 0.1% DMSO (Sigma) and 0.04% F-127 (AAT Bioquest) diluted in the culture medium for 1 h at 37 °C and 5% $CO_2$. The cells were subsequently washed with the culture medium. Before imaging, the culture medium was replaced with prewarmed artificial cerebrospinal fluid (ACSF) (10 mM HEPES, pH 7.4, 125 mM NaCl, 25 mM glucose, 2.5 mM KCl, 2 mM $CaCl_2$, 1 mM $MgCl_2$). Imaging was performed at 37 °C using a confocal microscope (LSM710, Carl Zeiss) equipped with a Plan-Apochromat 20x/0.8 objective lens. The dye-loaded olfactory interneurons were excited with a 488 nm argon laser, and the emission signals were detected simultaneously at 525 ± 25 nm and >611 nm. Differential interference contrast (DIC) and fluorescence images were acquired at 5-s intervals for 15 min or 30 min. To image cells expressing microRNA, EBFP fluorescence was used as an indicator of transfected cells. $[Ca^{2+}]_i$ analyses were performed using ImageJ software (NIH). The fluorescence intensities at 525 nm and 650 nm in the same region were measured over time at the soma. $[Ca^{2+}]_i$ was calculated as the ratio of the fluorescence intensity at 525 nm to the fluorescence intensity at 650 nm. The average of $[Ca^{2+}]_i$ was set as the baseline ($F_0$), and the $[Ca^{2+}]_i$ in each frame (F) was normalized to the baseline ($F/F_0$). A change in $[Ca^{2+}]_i$ was judged as a $Ca^{2+}$ transient if $F/F_0 > 40\%$.

## Mechanical stretching of the leading process

Neurons suspended in a mixture of 75% Matrigel and 25% L-15 medium were plated on elastic chambers and cultured for 2 days as

described above. For $Ca^{2+}$ imaging, after a control image was acquired, the chamber was set on the stretching device (Strex, catalog number STB-CH-04ST-20) and stretched uniaxially (20% stretch) (Appendix Fig. S3A). We then identified the same cell and acquired $Ca^{2+}$ images after the stretching process. We also measured the angle of the leading process with respect to the stretch direction ($\theta$) (Appendix Fig. S3B). When the angle was between 0° and 30°, we determined that the leading process was extended (parallel extension). On the other hand, when the angle was between 60° and 90°, the stretch direction was determined to be perpendicular to the leading process. For immunocytochemical analysis, olfactory interneurons were subjected to 20% uniaxial stretch for 15 min at 37 °C before fixation.

## Analyses of neuronal migration in Matrigel

Before observation, the culture medium was replaced with L-15 medium containing 2% B-27 supplement, 1 mM glutamine and 100 U/mL penicillin–streptomycin. Phase contrast images were acquired every 2 min for 2 h at 37 °C using a fluorescence microscope (IX81, Olympus) equipped with Plan Fluor 40x, 0.60 NA (Olympus) and an electron multiplying-CCD camera (EMCCD, iXon DU888, Andor) and MetaMorph. To image the cells expressing microRNA, mCherry fluorescence was used as an indicator of transfected cells. Time-lapse images were analyzed and quantified using ImageJ and the manual tracking plugin. The migration speed (Fig. 5E) was calculated by dividing the migration distance of the soma by the observation time. When the soma translocated any distance during the 2-min image acquisition intervals, we judged that neurons were in the somal translocation phase. The somal translocation speed (Fig. 5D) was calculated by dividing the migration distance of the soma during the somal translocation phase by the observation time.

## qPCR analysis of migrating neurons

SVZ explants were prepared and cultured as previously described (Minegishi et al, 2018). Briefly, the SVZs were dissected from the P5 mouse brains and cut into blocks 100–150 μm in diameter. The SVZ tissue blocks were then embedded in a mixture of 75% Matrigel and 25% L-15 medium, and cultured in neurobasal medium containing 2% B-27 supplement, and 1 mM glutamine in a humidified 5% $CO_2$ incubator at 37 °C. After 2 days in culture, the SVZ blocks were removed from the olfactory interneurons that migrated out from the blocks. Then, the total RNA of the migrating neurons was isolated using an RNeasy mini kit (Qiagen) according to the manufacturer's protocol followed by DNaseI digestion (Thermo Fisher Scientific, catalog number 11766051). The RNA was then reverse transcribed into cDNA using Superscript IV VILO master mix (Thermo Fisher Scientific) according to the manufacturer's protocol, and cDNA was used in the qPCR assays. qPCR assays were performed using the LightCycler 96 system (Roche), KAPA SYBR FAST qPCR kit (Nippon Genetics) and the following primer sets: Piezo1, 5′-TTGACCCTGCCAACTGGTTT-3′ and 5′-GCCTCAAACACCAGCAACAG-3′; Piezo2, 5′-GTATTGGATC-TACGTCTGCGG-3′ and 5′-CAGGA TCTTCCTCCACCACTCG-3′; TRPC1, 5′-CAGAAGGACTGTGT GGGCAT-3′ and 5′-GAACAGAGCAAAGCAGGTGC-3′; TRPC3, 5′-AGTGTCTGGTCGTGTTGGTC-3′ and 5′-AAGAATCTTCCCC

AGCCTGC-3′; TRPC5, 5′-TACCAATGTGAAGGCCCGAC-3′ and 5′-TATCAGCATGATCGGCAATG-3′; TRPC6, 5′-GCCGTCCAA ATCTCAGCCGTT-3′ and 5′-TGGACAGGAGCTGTTGCTGAC-3′; TRPV2, 5′-GTTTGACCGTGACCGACTCT-3′ and 5′-CGTCTT TCCAGTGGAGCCTT-3′; TRPV4, 5′-GCTTCTTCCAGCCCAAG-GAT-3′ and 5′-TGTCGCCTCATGTCAGCTTT-3′; TRPA1, 5′-ACAAGAAAGCCAGCCCTCTC-3′ and 5′-GGTTGCAGCAAA ATGGAGGG-3′; TACAN, 5′-CTGCAGCAGGACTTCCAAGGT-3′ and 5′-CAGTTGGCCTGGAGTTTGGTC-3′; Tmem63a, 5′-CA GGGCAACAAGACTTTGAA-3′ and 5′-TGTGCCTCTGAAAA-GACAGG-3′; Tmem63b, 5′-AGAACAGGACGACCATGCACA-3′ and 5′-AGAGCCAGCGGAAGAAGAGGT-3′; GAPDH, 5′-TG ATGGGTGTGAACCACGAG-3′ and 5′-GGCATGGACTGTGGT-CATGA-3′.

## Preparation of cell lysate and immunoblotting

Cultured neurons were lysed with RIPA buffer (50 mM Tris-HCl (pH 8.0), 1 mM EDTA, 150 mM NaCl, 1% Triton X-100, 0.1% SDS, 0.1% sodium deoxycholate, 1 mM DTT, 1 mM PMSF, and 0.01 mM leupeptin), and incubated for 2 h at 4 °C. The cell lysate was then centrifuged at $17,900 \times g$ for 10 min at 4 °C. The supernatant was mixed with an equal volume of 2 x SDS sample buffer (131 mM Tris-HCl, pH 6.8, 21% glycerol, 4% SDS, 12 M urea, 0.05% bromophenol blue and 5% β-mercaptoethanol). The mixture was incubated for 2 h at 37 °C, followed by SDS-polyacrylamide gel electrophoresis. Immunoblotting was performed as previously described (Minegishi et al, 2018). The following primary antibodies were used in immunoblotting: rabbit anti-Tmem63b (1:200) (Novus Biologicals, catalog number NBP2-58473), mouse anti-actin (1:10,000) (Millipore, catalog number MAB1501R). The following secondary antibodies were used for immunoblotting: HRP-conjugated donkey anti-rabbit IgG (1:2000) (GE Healthcare, catalog number NA934) and HRP-conjugated donkey anti-mouse IgG (1:5000) (Bio-Rad, catalog number 1706516).

## Immunocytochemical analysis

For Fig. 3D, cultured olfactory interneurons were fixed with 4% paraformaldehyde (PFA) dissolved in phosphate-buffered saline (PBS) for 15 min on ice. For Fig. 4A and Appendix Fig. S6B, olfactory interneurons subjected to 20% uniaxial stretch were fixed with 4% PFA dissolved in Krebs buffer (118 mM NaCl, 4.7 mM KCl, 1.2 mM KH$_2$PO$_4$, 1.2 mM MgSO$_4$, 4.2 mM NaHCO$_3$, 2 mM CaCl$_2$, 10 mM glucose, 400 mM sucrose, and 10 mM HEPES, pH 7.0) for 10 min on ice. The cells were treated with 0.1% Triton X-100 in PBS for 15 min on ice and 10% fetal bovine serum in PBS for 1 h at room temperature. They were then incubated with primary antibody diluted in PBS containing 10% fetal bovine serum overnight at 4 °C. The following primary antibodies were used: rabbit anti-Tmem63b (1:200) (Novus Biologicals, catalog number NBP2-58473), rabbit anti-myosin light chain 2 (1:500) (Abcam, catalog number ab79935) and mouse anti-phospho-myosin light chain 2 (Ser19) (1:500) (Cell Signaling Technology, catalog number 3675). Cells were washed with PBS, and then incubated with secondary antibody and 1 µg/mL DAPI (Thermo Fisher Scientific) diluted in PBS for 1 h at room temperature. The following

secondary antibodies were used: Alexa Fluor 647-conjugated donkey anti-rabbit IgG (1:1000) (Abcam, catalog number ab150075), Alexa Fluor 594-conjugated donkey anti-rabbit (1:1000, Jackson ImmunoResearch Laboratories, catalog number 711-585-152) and Alexa Fluor 488 conjugated goat anti-mouse (1:1000) (Thermo Fisher Scientific, catalog number A-11029). The immunostained cells were mounted with 50% (v/v) glycerol (Nacalai Tesque) in PBS. Fluorescence images were acquired using a confocal laser microscope (Stellaris 8, Leica) and imaging software (LAS X, Leica). For Fig. 3D, we used HC PL APO CS2 100x 1.40 NA objective lens (Leica). For Fig. 4A and Appendix Fig. S6B, we used an HC Plan-Apochromat 20x, 0.75 NA objective lens (Leica). To acquire images of the entire olfactory interneurons, we obtained 6–15 confocal images at 0.5 µm (for Fig. 3D) or 2 µm (for Fig. 4A and Appendix Fig. S6B) intervals in the Z axis direction. Then, the confocal images were volume-stacked, and the fluorescence signals were quantified using ImageJ.

## Immunohistochemistry

Postnatal days 5 mouse brains were fixed with 4% PFA for 12 h. PFA was replaced with 30% sucrose in PBS for overnight at 4 °C, and then frozen in Tissue Tek optimal cutting temperature (OCT, Sakura Finetek). Tissue blocks were cut into sagittal sections of 12 µm thickness using a cryostat (NX70, Micro-edge Instruments). The sections were incubated with blocking solution (10% normal goat serum (Vector Laboratories) and 0.3% Triton X-100 (Wako) in phosphate buffer) for 2 h at room temperature. After blocking, they were incubated with primary antibody diluted with the blocking solution for overnight at 4 °C. They were washed with phosphate buffer containing 0.3% Triton X-100 (PBT buffer) and incubated with secondary antibody and 1 µg/ml 4′,6-diamidino-2-phenylindole (DAPI) diluted with blocking solution overnight at 4 °C. They were then washed with PBT buffer and mounted with 50% glycerol (Nacalai) in PBS. The following primary antibodies were used: rabbit anti-Tmem63b (1:200) (Novus Biologicals, catalog number NBP2-58473). The secondary antibody used was Alexa Fluor 594-conjugated donkey anti-rabbit (1:1000, Jackson ImmunoResearch Laboratories, catalog number 711-585-152). Fluorescence images were acquired using a confocal laser microscope (BZ-X710, Keyence) equipped with a Plan-Apochromat 20x, 0.75 NA objective lens (Nicon) and imaging software (BZ-X viewer, Keyence).

## Chemicals

The following reagents were used at the indicated concentrations in this study: 5 µM GsMTx4 (Tocris Bioscience), 100 µM blebbistatin (Sigma), and 10 µM ML7 (Tocris Bioscience). A stock solution of GsMTx4 was prepared in H$_2$O. Stock solutions of blebbistatin and ML7 were prepared in DMSO.

## Quantification and statistical analysis

Statistical analyses were performed using Excel 2016 (Microsoft) or GraphPad Prism 7 (GraphPad Software). For samples with more than 7 data points, the D'Agostino–Pearson normality test was used to determine whether the data followed a normal distribution. For

cases in which the number of data points was between 3 and 7, the Shapiro–Wilk test was used for the normality test. We also tested the equality of variation with the F test for two independent groups that followed normal distributions. Significance tests were performed as follows: (1) two-tailed paired *t*-test to compare normally distributed data from two dependent groups; (2) two-tailed Wilcoxon signed-rank test to compare nonnormally distributed data from two dependent groups; (3) two-tailed unpaired Student's *t*-test to compare normally distributed data with equal variance from two independent groups; (4) two-tailed unpaired Welch's *t*-test to compare normally distributed data with unequal variance from two independent groups; (5) two-tailed Mann–Whitney *U*-test to compare nonnormally distributed data from two independent groups; (6) two-tailed one-way ANOVA with Tukey's post hoc test to compare normally distributed data from three groups; (7) two-tailed Kruskal–Wallis tests with Dunn's multiple comparison test to compare nonnormally distributed data from three groups; and (8) two-tailed Spearman's correlation coefficient test to analyze nonnormally distributed data. The statistical information and number of samples for each experiment are indicated in the figure legends. For detailed statistical results including the test statistics and exact p values, see the statistical source data associated with each figure. All data are shown as the mean ± SEM. Statistical significance was defined as $***p < 0.01$; $**p < 0.02$; $*p < 0.05$; ns, not significant. For the experiments using shootin1 KO mice, littermates were allocated into the experimental groups by genotyping. For the other experiments, sample allocation was randomized as cells were derived from the same resource. The numbers for the groups of each sample were based on those in previously published studies. No data were excluded if the experiments were successfully performed. All experiments were performed at least three times and reliably reproduced. Investigators were blind to the experimental groups for each analysis, except biochemical analyses.

## Materials availability

All unique materials generated in this study are available from the corresponding authors with a completed Materials Transfer Agreement.

# Data availability

The source data are provided in this study. The codes used for SICM data analyses have been deposited at Mendeley Data (doi: 10.17632/z8km7746tp.2) and are publicly available as of the date of publication. Any additional information required to reanalyze the data reported in this paper is available from the corresponding authors upon request.

The source data of this paper are collected in the following database record: biostudies:S-SCDT-10_1038-S44318-024-00326-8.

# Peer review information

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

## Acknowledgements

We thank Dr. Motohiro Nishida for valuable discussions; Mieko Ueda and Kazumi Maekawa for technical support; and Satoko Shimamura for kind encouragement. This research was supported in part by AMED under Grant Number JP17gm0810011 (NI), JSPS KAKENHI (JP19H03223, NI), JSPS Grants-in-Aid for Early-Career Scientists (JP19K16258 and JP23K14181, TM), and the Osaka Medical Research Foundation for Incurable Diseases (TM).

## Author contributions

**Takunori Minegishi**: Conceptualization; Resources; Data curation; Formal analysis; Funding acquisition; Investigation; Writing—original draft; Project administration; Writing—review and editing. **Honami Hasebe**: Formal analysis; Investigation. **Tomoya Aoyama**: Formal analysis; Investigation. **Keiji Naruse**: Supervision. **Yasufumi Takahashi**: Software; Supervision. **Naoyuki Inagaki**: Supervision; Funding acquisition; Writing—original draft; Project administration; Writing—review and editing.

Source data underlying figure panels in this paper may have individual authorship assigned. Where available, figure panel/source data authorship is listed in the following database record: biostudies:S-SCDT-10_1038-S44318-024-00326-8.

