## [Peer Review File · The EMBO Journal]

Mechanical signaling through membrane tension induces somal translocation during neuronal migration

Takunori Minegishi, Honami Hasebe, Tomoya Aoyama, Keiji Naruse, Yasufumi Takahashi, and Naoyuki Inagaki

Corresponding authors: Naoyuki Inagaki (ninagaki@bs.naist.jp) , Takunori Minegishi (t-minegishi@bs.naist.jp)

Review Timeline:

Submission Date:	16th May 24
Editorial Decision:	20th Jun 24
Revision Received:	15th Oct 24
Editorial Decision:	8th Nov 24
Revision Received:	14th Nov 24
Accepted:	18th Nov 24

Editor: Ieva Gailite

Transaction Report:

Dear Dr. Inagaki,

Thank you for submitting your manuscript for consideration by the EMBO Journal. We have now received comments from three reviewers, which are included below for your information.

As you can see, all reviewers find the proposed role of the mechanosensing channel TMEM63b in regulation of neuronal migration of interest. However, they also point out several aspects that they find need strengthening before they can support publication here. Based on the overall interest expressed in all reports, I would like to invite you to revise the manuscript in response to referee concerns.

In particular, reviewer #1 asks for further support to the proposed role of membrane tension in activation of TMEM63B, and indicates that TMEM63B localisation needs to be shown to conclude about its potential role at the plasma membrane. Since this is an important and central part of the proposed mechanism, please consider alternative approaches to substantiate the role of membrane tension in this process. Furthermore, reviewer #2 asks for some evidence of the involvement of TMEM63B and membrane tension in neuronal migration in a more in vivo-like environment, e.g., in brain slices, via application of the Flipper-TR probe. I think that testing Tmem63b knockdown in such context would also enhance the impact of the manuscript. Finally, reviewer #3 asks for better quantification and data presentation. From the editorial side, I appreciate that the analysis of a potential osmotic gradient along olfactory interneuron migration path as requested by reviewer #2 might be challenging to show and will not be required for acceptance of the manuscript.

We generally allow three months as standard revision time, which I have now extended to five months. Should you foresee a problem in meeting this deadline, please let me know in advance to discuss an extension. I should add that it is The EMBO Journal policy to allow only a single major round of revision and that it is therefore important to resolve the main concerns at this stage. I think that it would be useful to discuss the revision in more detail via email or phone/videoconferencing - please let me know which option you prefer.

As a matter of policy, competing manuscripts published during this period will not negatively impact on our assessment of the conceptual advance presented by your study. However, please contact me as soon as possible upon publication of any related work to discuss the appropriate course of action.

When preparing your letter of response to the referees' comments, please bear in mind that this will form part of the Review Process File and will therefore be available online to the community. For more details on our Transparent Editorial Process, please visit our website: <https://www.embopress.org/page/journal/14602075/authorguide#transparentprocess>. Please also see the attached instructions for further guidelines on preparation of the revised manuscript.

Please feel free to contact me if you have any further questions regarding the revision. Thank you for the opportunity to consider your work for publication. I look forward to discussing your revision.

With best regards,

Ieva

- a point-by-point response to the referees' comments, with a detailed description of the changes made (as a word file).
 - a word file of the manuscript text.
 - individual production quality figure files (one file per figure)
 - a complete author checklist, which you can download from our author guidelines (<https://www.embopress.org/page/journal/14602075/authorguide>).
 - Expanded View files (replacing Supplementary Information)
- Please see out instructions to authors
<https://www.embopress.org/page/journal/14602075/authorguide#expandedview>

We realize that it is difficult to revise to a specific deadline. In the interest of protecting the conceptual advance provided by the work, we recommend a revision within 3 months (18th Sep 2024). Please discuss the revision progress ahead of this time with the editor if you require more time to complete the revisions.

Referee #1:

Minegishi et al. report that somal translocation during neuronal migration involves mechanical signaling through the TMEM63b channel.

The manuscript is clearly written and the evidence for mechanical signaling is strong. However, there are a few major issues that should be addressed:

- 1) It is not convincing that the mechanical signal is membrane tension. The mechanical measurements using SICM for most of the paper are of Young's modulus, not membrane tension.

In Fig. 1A-C, the fluorescence lifetime of FlipperTR is measured. Although Flipper-TR lifetime does depend on membrane tension, it is also influenced by membrane order (e.g. see Amos...Tamm, MBoC, 2024), and possibly other factors. In addition, the differences in Flipper-TR signals between growth cones and the somatas are not very large, and could easily be influenced by how the data is processed.

In this respect, there is not enough information about the processing of the FLIM data. What regions of the cell are considered for the analysis? In B, are all process lengths averaged together? How did you define the different regions? A double exponential fit is used for extracting lifetimes, but which lifetime is plotted in Fig 1A-C? If you change the cutoffs for the fits, how the analysed regions are defined, etc., how do the results change?

I suggest not insisting that the mechanical signaling is via membrane tension, since this part is not on very solid ground.

- 2) The literature must be discussed better; there are reports that contradict or support the current findings.

- Dai and Sheetz (Cell, 1995, 83:693) reported that there is a membrane tension gradient in growing chick DRG axons, with lower tension in the growth cone than in the soma. This gradient drives membrane flow from the growth cone to the cell body. Importantly, the membrane tension was found to be very low, likely not enough to activate any mechanosensitive channels.

Popov et al found membrane flow in the opposite direction in Xenopus neurites (Popov, S., Brown, A., and Poo, M. 1993, Science 259:244-246), suggesting the tension is higher at the tip than the soma...

Recently, Shi, Innes-Gold, and Cohen reported that tension propagates rapidly along axons, allowing long-range signaling (Sci. Adv. 8:eabo1297, 2022).

Similarly, Gomis Perez...Karatekin reported that membrane tension propagates very rapidly within synaptic terminals, and can be transmitted to the axon (Sci. Adv. 8:eabl4411, 2022).

The last two reports seem to support the idea that membrane tension can rapidly be communicated from the growth cone/terminal to the soma. However, another important question is: what is the increase in tension sensed at the soma that could support the opening of the TMEM63 channels? Some discussion of these points is needed.

3) Where is TMEM63?

At the soma? The authors should at least do immunofluorescence. Do calcium entry sites and TMEM63 co-localize?

TMEM63 was recently reported to be present in lysosomes (Li...Jan, Nature Cell Biol. 2024, 26:393). Both mouse TMEM63A and B showed good colocalization with the lysosomal marker LAMP-2 in Neuro-2a cells.

If TMEM63 is not at the plasma membrane, it cannot respond to increased plasma membrane tension.

4) GsMTx4 inhibits Piezo and TRP mechanosensitive channels. Is there evidence it is also effective in inhibiting TMEM63?

5) A possible technical issue: would SICM report higher apparent stiffness for cell regions that are thinner (such as the growth cone)? Is there a way to test this? (e.g. grow the cells on a soft substrate?).

In other words, is the higher stiffness found at the growth cone a technical artifact, because the growth cone can only deform very little due to its limited thickness?

Minor issues:

6) Fig 4 B vs C. In C, the leading process lengths are ~0.5 um. Is this leading process migration speed instead???

Referee #2:

General assessment

This manuscript provides a compelling mechanical model for the migration of mouse olfactory interneurons. The authors showed that by extending their leading process, the olfactory interneurons trigger plasma membrane tension, which precedes their somal translocation. This membrane tension activates the mechanosensitive channel Tmem63b that trigger intracellular Ca²⁺ oscillations. The authors showed that silencing Tmem63b channels impairs somal translocation during migration by preventing tension-triggered Ca²⁺ currents, thereby reducing myosin II activity via reduction of MLCK phosphorylation. The manuscript provide important finding with a significant impact in the field of both mechanobiology and neuronal migration. However, the manuscript can be improved as several issues should be addressed.

Major comments

Figure 1. How is the membrane tension regulation of olfactory interneuron migration stands in vivo/in situ?

> The authors should show the distribution of Tmem63b at mRNA or protein level in the SVZ/RMS and the olfactory bulb.

> Here, the extension of the leading process boosts membrane tension regardless of whether interneurons migrate in a 3D or a 2D environment. Previous work on retinal cells showed the extension of their leading process during migration is also influenced by the extracellular environment, which is distinct in 2D and 3D. It is thus important to assess olfactory interneuron membrane tension upon extension of their leading process while migrating in vivo or in situ (brain organotypic slices), where environmental mechanical cues are preserved.

> Tmem63b is a mechanosensitive channel that can also be activated by osmotic stimuli. The reviewer is wondering whether possible osmotic changes along the migration path of these interneurons may also contribute to their movement? The authors should clarify this point and discuss it.

Figure 3B. mRNA levels from different genes cannot be compared by qPCR since the efficiency of the primers is not comparable. Authors should remove the statistics from the figure.

Figure S4. Here, the authors suggest that increase in membrane tension triggers Ca²⁺ bursts, which in turn activates actomyosin cytoskeleton to generate contractile forces and move the soma forward. According to the cage organization of microtubules (MTs) around the nucleus and their role in somal translocation (at least in some types of migrating neurons), the reviewer is wondering whether MTs also respond to Tmem63b activation? Do membrane tension changes and activation of Tmem63b also affect the growth/stability of MTs? How are actin and MT dynamics coordinated?

Minor comment

> Figure S3. The western blot picture is not convincing. Please replace it.

> The following paper should be included in the discussion: Stretch-activated ion channel TMEM63B associates with

Referee #3:

In this study, the authors explore the critical link between leading process extension and somal translocation in neuronal migration. The authors used extensive advanced and live cell microscopy to demonstrate that the extension of the leading process increases plasma membrane tension, which subsequently activates the mechanosensitive ion channel Tmem63b, triggering Ca²⁺ influx and actomyosin activation at the cell rear. This mechanical signaling pathway is essential for generating the robust forces needed to translocate the soma against the mechanical barriers of the three-dimensional environment, thereby facilitating rapid and saltatory neuronal migration. The study is overall of importance, particularly providing valuable insights into the cellular and molecular processes underlying neuronal migration, with potential implications for understanding and addressing migration-related neurological disorders. There are several points to be addressed by the authors:

1. In Figure 2B, F, H, and I, only traces from a single cell are presented. The authors should consider including additional traces either in this figure or in a supplementary figure to demonstrate the robustness of these observations.
2. The finding that perpendicular stretching did not affect calcium influx is quite interesting. The authors should discuss the potential reasons behind this observation.
3. Do the authors refer to microRNA or siRNA when they mention miRNA? Clarification is needed.
4. In Figure 4A and B, there is a sudden retraction of the leading process at 120'. Is this phenomenon consistently observed?
5. In Figure 4B, only one trace is presented. The authors should include more representative traces and an average of all cells observed to provide an overall view of the process.
6. The authors should discuss potential similarities between olfactory neuronal migration and the migration of cortical neurons to expand the impact of this study. They may reference Vallee RB et al., Trends in Cell Biol 2009; Metin et al., Eur J Neurosci 2006; and Jheng GW et al., Biochem Biophys Res Commun 2018.

Responses to the comments of Referee #1
(Manuscript Number: EMBOJ-2024-117904).

We thank Referee #1 for a thorough review and constructive suggestions. The following are our point-by-point responses to the comments. We have added new data (revised Figs. 3C-G, 5B, EV1, EV2, EV4 and EV7); the original Figure 3 and 4 have been separated into two Figures. Accordingly, the number of the Figures has been increased from four to six. Changes in the text of the manuscript are highlighted in red font.

Major issues:

1) It is not convincing that the mechanical signal is membrane tension. The mechanical measurements using SICM for most of the paper are of Young's modulus, not membrane tension.

Response 1-1: We agree with the referee's comment. As described in the original text (lines 88-90), SICM has been used to monitor plasma membrane tension through cell stiffness measurements. Indeed, a number of studies have used membrane stiffness as an indicator of plasma membrane tension (Mandriota *et al.*, *Nat Mater* 18, 1071-1077, 2019; Barger *et al.*, *Nat Commun* 10, 1249, 2019; Bednarska *et al.*, *PNAS* 117, 21637-21646, 2020; Lachowski *et al.*, *ACS Nano* 16, 4322-4337, 2022), because an increase in the plasma membrane tension increases membrane stiffness.

As a key factor, the actin cortex underlying the plasma membrane may also affect the membrane stiffness (Rheinlaender and Schaffer, *Soft Matter* 9, 3230-3236, 2013). In response to the referee's, comment we have described this limitation of the measurement in the revised text (lines 92-93). In this regard, we measured the membrane stiffness at the same proximal regions of the leading process (Fig. 1D). By monitoring the change in stiffness of the same regions, we minimized the possible influence of differences in the local actin cortex. Nevertheless, as we cannot rule out the possibility that other factors could influence the stiffness data, we also used another method to assess the reliability of the data (see **Response 1-6**).

In Fig. 1A-C, the fluorescence lifetime of FlipperTR is measured. Although Flipper-TR lifetime does depend on membrane tension, it is also influenced by membrane order (e.g. see Amos...Tamm, MBoC, 2024), and possibly other factors.

Response 1-2: As pointed out by the referee, the fluorescence lifetime of Flipper-TR is affected by the order of the lipid membrane. In response to this comment, we have described this limitation of plasma membrane tension measurement using Flipper-TR (lines 84-85). In Fig. 1A-B, we have analyzed the same cell type bearing the leading process of different lengths under the same condition. In addition, a statistically sufficient number of neurons were analyzed, so we consider that the influences of membrane order and other factors are minimized and that the obtained fluorescence lifetimes detect the difference in the plasma membrane tension. We also used another

method to assess the reliability of the data (see **Response 1-6**).

In this respect, there is not enough information about the processing of the FLIM data. What regions of the cell are considered for the analysis? In B, are all process lengths averaged together? How did you define the different regions?

Response 1-3: We are grateful for these comments, which helps to clarify our data analyses. In the revised version, we have described how we defined the cell regions in the Flipper-TR analyses more in detail (Fig. EV1). We defined the tip of the leading process with a palm-like morphology bearing filopodia and lamellipodia as the growth cone, and the remaining regions of the cell as the leading process shaft and soma (see **Data 1A for Referee #1**). For the Flipper-TR analysis of the growth cone, we drew an outline of the growth cone (red line, **Data 1B for Referee #1**) and measured the averaged fluorescence lifetime within the region (white lines). For the analysis of the leading process shaft and soma, a 0.5 μm wide line was drawn along the plasma membrane region (red line, **Data 1C for Referee #1**) and the averaged fluorescence lifetime was measured on the line. The original Fig. 1B shows the fluorescence lifetime at the leading process shaft plus soma and at the growth cone calculated by this method

(n = 129 cells).

A double exponential fit is used for extracting lifetimes, but which lifetime is plotted in Fig 1A-C? If you change the cutoffs for the fits, how the analysed regions are defined, etc., how do the results change?

Response 1-4: A double exponential fit was performed to extract lifetimes: as previously described (Colom *et al.*, *Nat Chem* 10, 1118-1125, 2018), the longer lifetime was used for the plots in Fig. 1A-C. We did not perform any cutoff for the fits. In the revised version, we have described the detailed information about the processing of the FLIM data in the Methods (lines 616-620).

In addition, the differences in Flipper-TR signals between growth cones and the somatas are not very large, and could easily be influenced by how the data is processed.

Response 1-5: As pointed out, the differences in Flipper-TR signals between growth cones and the soma are not very large. In this paper, the critical point of the data analysis is to examine the correlation between the leading process length and the membrane tension of the leading process shaft and soma (**Data 1C for Referee #1**), not to examine the difference between the membrane tension of the growth cone (**Data 1B for Referee #1**) and that of the leading process shaft and soma (**Data 1C for Referee #1**). In addition, the method of defining the measured areas in the leading process shaft plus soma and in the growth cone is not identical (see **Response 1-3**). We have therefore deleted the original Fig. 1B.

I suggest not insisting that the mechanical signaling is via membrane tension, since this part is not on very solid ground.

Response 1-6: We agree that plasma membrane tension measured by SICM or Flipper-TR may be influenced by other factors. Nevertheless, by using these different methods and minimizing the possible influence of other factors we obtained consistent data that leading process extension increases plasma membrane tension of migrating olfactory interneurons (**Responses 1-1 and 1-2**). Furthermore, using cell-stretching assays, we demonstrated that leading process extension activates the ion channel Tmem63b that is activated by the increase in plasma membrane tension (Murthy *et al.*, *eLife* 7, e41844, 2018). These data collectively support our conclusion that plasma membrane tension mediates the mechanical signaling. In response to the referee's comments, we have described these points in the revised manuscript (lines 84-87, 92-96, 148-151). In addition, we have toned down our conclusion, by changing "...indicate ..." to "...suggest ..." (lines 29, 65, 99).

2) The literature must be discussed better; there are reports that contradict or support the current findings.

- Dai and Sheetz (Cell, 1995, 83:693) reported that there is a membrane tension gradient in growing chick DRG axons, with lower tension in the growth cone than in the soma. This gradient drives membrane flow from the growth cone to the cell body. Importantly, the membrane tension was found to be very low, likely not enough to activate any mechanosensitive channels. Popov et al found membrane flow in the opposite direction in Xenopus neurites (Popov, S., Brown, A., and Poo, M. 1993, Science 259:244-246), suggesting the tension is higher at the tip than the soma...

Recently, Shi, Innes-Gold, and Cohen reported that tension propagates rapidly along axons, allowing long-range signaling (Sci. Adv. 8:eabo1297, 2022). Similarly, Gomis Perez...Karatekin reported that membrane tension propagates

very rapidly within synaptic terminals, and can be transmitted to the axon (Sci. Adv. 8:eabl4411, 2022). The last two reports seem to support the idea that membrane tension can rapidly be communicated from the growth cone/terminal to the soma.

Response 2-1: We thank the referee for reminding us these reports. Our data showed that the stretch along the leading process shaft, not perpendicular stretching, triggered Ca^{2+} influx through Tmem63b (Fig. 2H-J). The perpendicular stretching stretches the growth cone and the soma. On the other hand, the parallel stretching stretches the leading process in addition to the growth cone and soma. Thus, our data indicate that the stretch along the leading process shaft, rather than that at the growth cone and the soma, plays the main role in Tmem63b activation. Indeed, changes in leading process length are positively correlated with changes in plasma membrane tension along the leading process (Fig. 1) and Tmem63b is widely localized along the leading process **Data 2 for Referee #1**). Furthermore, the tension propagation mentioned by the referee would facilitate the propagation of the mechanical signaling. We have discussed these points in the revised manuscript, citing the above literature (lines 181-193).

However, another important question is: what is the increase in tension sensed at the soma that could support the opening of the TMEM63 channels?

Response 2-2: As explained above (**Response 2-1**), the leading process extension promoted a global increase in plasma membrane tension across the leading process shaft and soma (Fig. 1A,D), consistent with the above-mentioned report that the increase in tension can easily propagate to the soma. Indeed, leading process extension provoked Ca^{2+} transient widely in the neurons (**Data 3A for Referee #1**) as reported (Martini and Valdeomillos, *J Neurosci* 30, 8660-8670, 2010). On the other hand, the downstream effector actomyosin is localized at the rear of the soma (**Data 3B for Referee #1**). Thus, the global activation of Tmem63b would trigger a local actomyosin activation at the rear of the soma. We have discussed these issues in the revised manuscript (lines 213-216).

3) *Where is TMEM63? At the soma? The authors should at least do immunofluorescence. Do calcium entry sites and TMEM63 co-localize? TMEM63 was recently reported to be present in lysosomes (Li...Jan, Nature Cell Biol. 2024, 26:393). Both mouse TMEM63A and B showed good colocalization with the lysosomal marker LAMP-2 in Neuro-2a cells. If TMEM63 is not at the plasma membrane, it cannot respond to increased plasma membrane tension.*

Response 3: In response to the referee's question, we examined the localization of endogenous Tmem63b in olfactory interneurons using an anti-Tmem63b antibody. As shown in **Data 2 for Referee #1**, Tmem63b is localized at the plasma membrane (arrowheads) as well as in the cytoplasm (presumably in lysosomes). The localization of Tmem63b at the plasma membrane is consistent with our conclusion that Tmem63b senses an increase in plasma membrane tension and triggers Ca^{2+} signaling in olfactory interneurons. The global Ca^{2+} signal corresponds to the global Tmem63b localization (**Data 2 and 3A for Referee #1**). We described these data in the revised text (Fig. 3D) (lines 140-143).

4) *GsMTx4 inhibits Piezo and TRP mechanosensitive channels. Is there evidence it is also effective in inhibiting TMEM63?*

Response 4: As pointed out, GsMTx4 inhibits various mechanosensitive channels, including Piezo and TRP channels (Suchyna, *Prog Biophys Mol Biol* 130, 244-253, 2017), but there are currently no reports showing that GsMTx4 inhibits Tmem63b. To clarify this, we have added the description "Although there are currently no reports showing that GsMTx4 inhibits Tmem63b, we focused on this ion channel because of its high expression." in the revised text (lines 135-137). We speculate that other mechanosensitive channels, including Piezo and TRPC channels, may also mediate the tension-induced mechanical signaling for somal translocation. Indeed, olfactory

interneurons express several mechanosensitive channels, including Piezo and TRPC channels (Fig. 3B), and their inhibitor GsMTx4 (Gnanasambandam *et al.*, 2017; Suchyna, 2017) partially inhibits Ca²⁺ transients in these cells (Fig. 3A). We have discussed this possibility in the revised text (line 235-239).

5) A possible technical issue: would SICM report higher apparent stiffness for cell regions that are thinner (such as the growth cone)? Is there a way to test this? (e.g. grow the cells on a soft substrate?). In other words, is the higher stiffness found at the growth cone a technical artifact, because the growth cone can only deform very little due to its limited thickness?

Response 5: We agree with the referee's concern. As described in **Response 1-5** above, the critical point of this study is the correlation between leading process length and the membrane tension of the leading process shaft and soma. Since the quantification of membrane stiffness at the growth cone is not critical in our study, we have removed the description "the stiffness was higher at the growth cone)" in the revised text and deleted the original Fig. 1B.

Minor issues:

6) Fig 4 B vs C. In C, the leading process lengths are ~0.5 um. Is this leading process migration speed instead???

Response 6: We thank the referee for pointing out our mistake on the values of the y-axis. We have corrected them in the revised version (revised Fig. 5C).

Responses to the comments of Referee #2 and the changes in the revised version (Manuscript Number: EMBOJ-2024-117904).

We thank Referee #2 for a kind and thoughtful review with suggestions to improve our paper. The following are our point-by-point responses to the comments. We have added new data (revised Figs. 3C-G, 5B, EV1, EV2, EV4 and EV7); the original Figure 3 and 4 have been separated into two Figures. Accordingly, the number of the Figures has been increased from four to six. Changes in the text of the manuscript are highlighted in red font.

Major comments

Figure 1. How is the membrane tension regulation of olfactory interneuron migration stands in vivo/in situ?

> The authors should show the distribution of Tmem63b at mRNA or protein level in the SVZ/RMS and the olfactory bulb.

Response 1-1: According to the referee's comment, we examined the distribution of Tmem63b in the migratory pathway of olfactory interneurons by co-staining with anti-Tmem63b antibody and DAPI (**Data 1 for Referee #2**). The subventricular zone (SVZ, asterisk), rostral migratory stream (RMS, arrowheads) and olfactory bulb (OB) were visualized by DAPI. We detected Tmem63b expression in the migratory pathway of olfactory interneurons, including the SVZ, RMS and OB. We have added these data in the revised manuscript (Fig. 3C) (lines 137-140).

Data 1 for Referee #2. Distribution of Tmem63b in the migratory pathway of olfactory interneurons. A sagittal section of postnatal day 5 mouse brain labelled with anti-Tmem63b antibody and DAPI. Scale bar, 200 μ m.

> Here, the extension of the leading process boosts membrane tension regardless of whether interneurons migrate in a 3D or a 2D environment. Previous work on retinal cells showed the extension of their leading process during migration is also influenced by the extracellular environment, which is distinct in 2D and 3D. It is thus important to assess olfactory interneuron membrane tension upon extension of their leading process while migrating in vivo or in situ (brain organotypic slices), where environmental mechanical cues are preserved.

Response 1-2: In response to the referee's suggestion, we tried to monitor the plasma membrane tension of migrating olfactory interneurons *in vivo* using Flipper-TR. However, this was difficult because the Flipper-TR signal of individual leading processes was not identifiable in the brain slice (**Data 2 for Referee #2**). In addition, SICM cannot detect membrane tension of neurons within brain tissue.

Data 2 for Referee #2. Fluorescence images of a sagittal section of postnatal day 5 mouse brain stained with 1 μ M Flipper-TR. Arrowheads indicate the rostral migratory stream. The color bar indicates the lifetime in nanoseconds (ns). An enlarged view of the region in the box is shown on the right. Scale bars, 100 μ m (left); 20 μ m (right).

Therefore, we introduced another plasma membrane tension reporter MscL-cpGFP (Hsu *et al.*, *ACS Sens* 8, 12-18, 2023). This tension reporter contains the endoplasmic reticulum export signal of Kir2.1 at its C-terminus, which facilitates its plasma membrane localization. However, we could not observe its plasma membrane localization in cultured olfactory interneurons and HEK293T cells (**Data 3 for Referee #2**) (mCherry-CAAX is a plasma membrane marker).

Data 3 for Referee #2. Fluorescence images of an olfactory interneuron and a HEK293T cell co-expressing MscL-cpGFP and mCherry-CAAX (plasma membrane marker). Scale bars, 10 μ m.

So, we further fused a plasma membrane targeting signal, the CAAX tag, with a 20-amino acid long flexible linker peptide (GGGGS)₄ into the C-terminus of MscL-cpGFP (MscL-cpGFP-CAAX). As shown in **Data 4A for Referee #2**,

MscL-cpGFP-CAAX localized at the plasma membrane of HEK293T cells (TMD-mCherry is a plasma membrane marker).

To test tension sensitivity of MscL-cpGFP-CAAX, we performed the cyclic osmotic pressure test on HEK293T cells expressing MscL-cpGFP-CAAX and TMD-mCherry, where mCherry was fused to the transmembrane domain of transferrin receptor for membrane localization. We expected a decrease in the fluorescence ratio (MscL-cpGFP-CAAX/TMD-mCherry) with increasing plasma membrane tension. In this experiment, the osmolarity was repeatedly switched between iso-osmotic (\sim 330 mOsm) and hypo-osmotic (\sim 165 mOsm) conditions to generate alternating resting and elevated membrane tension (Hsu *et al.*, *ACS Sens* 8, 12-18, 2023). However, we could not observe changes in fluorescence intensity of MscL-cpGFP-CAAX in response to the increase in membrane tension (**Data 4A and B for Referee #2**), indicating that it is difficult to obtain reliable plasma membrane tension data using MscL-cpGFP-CAAX.

Thus, it is challenging to monitor the neuronal membrane tension in tissues: to our knowledge, no report has succeeded in doing so. Nevertheless, we have obtained consistent data that leading process extension increases the membrane tension *in vitro*, using two different methods (Flipper-TR and SICM). Furthermore, migrating olfactory

interneurons repeat dynamic leading process extension and somal translocation in the brain (O'Rourke *et al.*, *Science* 258, 299-302, 1992). Therefore, we believe that the mechanical signaling via plasma membrane tension plays a critical role *in vivo*. As the referee points out, the analysis of the membrane tension of migrating neurons *in vivo* is an important issue. We have discussed this in the revised text (lines 199-200).

> *Tmemb63b is a mechanosensitive channel that can also be activated by osmotic stimuli. The reviewer is wondering whether possible osmotic changes along the migration path of these interneurons may also contribute to their movement? The authors should clarify this point and discuss it.*

Response 1-3: As the referee pointed out, Tmem63b has also been reported to be activated by osmotic stimuli (Murthy *et al.*, *eLife* 7, 2018, e41844; Du *et al.*, *Cell Reports* 31, 2020, 107596). We agree with the referee's notion that possible osmotic changes along the migration pathway may also contribute to the movement of olfactory interneurons, since hypo-osmotic conditions increase membrane tension. We have discussed this possibility in the revised text (lines 194-200).

Figure 3B. mRNA levels from different genes cannot be compared by qPCR since the efficiency of the primers is not comparable. Authors should remove the statistics from the figure.

Response 2: We thank the referee for pointing out this. We have removed the statistical analysis from the qPCR data in the revised version (Fig. 3B).

Figure S4. Here, the authors suggest that increase in membrane tension triggers Ca²⁺ bursts, which in turn activates actomyosin cytoskeleton to generate contractile forces and move the soma forward. According to the cage organization of microtubules (MTs) around the nucleus and their role in somal translocation (at least in some types of migrating neurons), the reviewer is wondering whether MTs also respond to Tmem63b activation? Do membrane tension changes and activation of Tmem63b also affect the growth/stability of MTs? How are actin and MT dynamics coordinated?

Response 3: To our knowledge, there is no report showing that Tmem63b activation or Ca²⁺ transient affects the growth and stability of perinuclear microtubules in migrating neurons. It is also unclear how the actin and MT dynamics around the nucleus are coordinated. We have mentioned these points in the revised text (lines 216-223).

Minor comment

> *Figure S3. The western blot picture is not convincing. Please replace it.*

Response 4: Following the referee's comment, we replaced the original western blot picture with a new one (**Data 5A for Referee #2**) (Fig. EV6A). Since Tmem63b is a glycosylated membrane protein, it is detected as a broad band by western blot, as

reported by Zheng *et al.* (*Neuron* 111, 3195-3210, 2023) (**Data 5B for Referee #2**).

> The following paper should be included in the discussion: Stretch-activated ion channel TMEM63B associates with developmental and epileptic encephalopathies and progressive neurodegeneration (PMID: 37421948).

Response 5: We thank the referee for this suggestion. We have added the following discussion in the revised text: “Defects in neuronal migration cause brain diseases, including brain malformations, epilepsy, and intellectual disability (Ross & Walsh, 2001; Valiente & Marin, 2010; Evsyukova *et al.*, 2013; Stouffer *et al.*, 2016; Nakajima *et al.*, 2021). Recently, heterozygous mutations of *TMEM63b* have been identified in patients with severe early-onset developmental and epileptic encephalopathies and progressive neurodegeneration (Vetro *et al.*, 2023). These patients exhibit brain malformations, including multiregional or widespread white matter, dysmorphic lateral ventricles, thinning of the corpus callosum, cerebellar atrophy, and atrophy of the cerebral cortex (Vetro *et al.*, 2023). Cells expressing the mutant Tmem63b have a reduced response to membrane stretch (Vetro *et al.*, 2023), suggesting a possible linkage between the neuronal migration defects caused by Tmem63b dysregulation and brain malformations.” (lines 251-261).

Responses to the comments of Referee #3 and the changes in the revised version (Manuscript Number: EMBOJ-2024-117904).

We thank Referee #3 for a careful and constructive review to improve our paper. The following are our point-by-point responses to the comments. We have added new data (revised Figs. 3C-G, 5B, EV1, EV2, EV4 and EV7); the original Figure 3 and 4 have been separated into two Figures. Accordingly, the number of the Figures has been increased from four to six. Changes in the text of the manuscript are highlighted in red font.

1. In Figure 2B, F, H, and I, only traces from a single cell are presented. The authors should consider including additional traces either in this figure or in a supplementary figure to demonstrate the robustness of these observations.

Response 1: According to the suggestion, we have shown additional traces of the original Fig. 2B, F, H, and I in Figs. EV2 and EV4 of the revised manuscript.

2. The finding that perpendicular stretching did not affect calcium influx is quite interesting. The authors should discuss the potential reasons behind this observation.

Response 2: As pointed out, our data showed that parallel stretching, not perpendicular stretching, triggered Ca^{2+} influx through Tmem63b. Perpendicular stretching stretches the growth cone and the soma. On the other hand, the parallel stretching stretches the leading process in addition to the growth cone and the soma. Therefore, our data suggest that the stretching along the shaft of the leading process is the key step that activates Tmem63b. Consistently, Tmem63b is widely localized along the leading process (**Data 1 for Referee #3**) and changes in leading process length are positively correlated with changes in plasma membrane tension along the leading process (Fig. 1). We have discussed about this point in the revised text (lines 181-189).

Data 1 for referee #3. Localization of endogenous Tmem63b in olfactory interneurons. An olfactory interneuron immunostained by anti-Tmem63b and anti-doublecortin antibodies. Doublecortin antibody was used as a marker of olfactory interneuron. Scale bars, 10 μm (left); 2 μm (right).

3. Do the authors refer to microRNA or siRNA when they mention miRNA? Clarification is needed.

Response 3: According to the referee's comment, we have changed the word "miRNA" to "microRNA" in the revised manuscript.

4. In Figure 4A and B, there is a sudden retraction of the leading process at 120'. Is this phenomenon consistently observed?

Response 4: In response to this comment, we have re-examined the changes in the leading process length of olfactory interneurons expressing Tmem63b microRNA. The sudden retraction of the leading process was not consistently observed. To avoid misinterpretation, we have replaced this figure with a more typical one in the revised text.

5. In Figure 4B, only one trace is presented. The authors should include more representative traces and an average of all cells observed to provide an overall view of the process.

Response 5: According to the referee's comment, we have added more representative traces and an average of all cells observed in the revised manuscript (Figs. 5B and EV7).

6. The authors should discuss potential similarities between olfactory neuronal migration and the migration of cortical neurons to expand the impact of this study. They may reference Vallee RB et al., Trends in Cell Biol 2009; Metin et al., Eur J Neurosci 2006; and Jheng GW et al., Biochem Biophys Res Commun 2018.

Response 6: We thank the referee for this suggestion to improve our paper. We have added the following discussion to explain the potential similarities between the migrations of olfactory interneurons and cortical neurons, as well as cerebellar granular cells and gonadotropin-releasing hormone-expressing neurons: "Neurons migrate by repeating leading process extension and somal translocation. Ca²⁺ transients are required for somal translocation not only of olfactory interneurons but also of cortical neurons, cerebellar granule cells, and gonadotropin-releasing hormone-expressing neurons (Komuro & Rakic, 1996; Martini & Valdeolmillos, 2010; Hutchins *et al*, 2013; Rash *et al*, 2016). Furthermore, somal translocation of these neurons is driven by actomyosin and dynein motor complexes (Schaar & McConnell, 2005; Metin *et al*, 2006; Tsai *et al*, 2007; Solecki *et al.*, 2009; Vallee *et al*, 2009; Martini & Valdeolmillos, 2010; Hutchins *et al.*, 2013; Jheng *et al*, 2018). Therefore, the present mechanical signaling could underlie the cell migration of a wide range of neuronal types."(lines 240-248).

Dear Dr. Inagaki,

Thank you for submitting a revised version of your manuscript. We have now received input from two of the original reviewers, who find that most of previous concerns have been addressed satisfactorily and recommend acceptance of the manuscript after a minor textual revision as requested by reviewer #1.

Additionally, there remain a few editorial points that need addressing before I can extend official acceptance of the manuscript:

1. Please reduce the number of keywords to five.
2. In the Author Checklist file, please check the information in the row 52, "Cell lines", as it is currently ambiguous.
3. Please make sure that the order of the sections in the manuscript is as follows: Abstract / Keywords / Introduction / Results / Discussion / Methods / Acknowledgments / Disclosure and competing interests statement / References / Figure legends / Tables and their legends / Expanded View Figure legends.
4. CRedit has replaced the traditional author contributions section because it offers a systematic, machine-readable author contributions format that allows for more effective research assessment. Please remove the Authors Contributions from the manuscript and use the free text boxes beneath each contributing author's name in our online submission system to add specific details on the author's contribution. More information is available in our guide to authors.
5. Please remove movie legends from the manuscript text file and zip together with each movie file. Further information is available here: <https://www.embopress.org/page/journal/14602075/authorguide#expandedview>
6. Please correct the name of the "Supplemental Information" file to "Appendix", add page numbers to the table of contents and remove the list of other supplemental information from the file. Please rename the figures "Appendix Figure S1" - S6 and update their callouts in the manuscript text.
7. In our standard image integrity check, we noted a reuse of the image panels between the panels in Fig. 1A and EV1A. If the reuse is intentional, please indicate this in the figure legends.
8. Source data for Figures 5D and 5E appear identical, please check.
9. In our standard source data check, we have noted unexplained numerical duplications in the source data for figures 2E, 3B, EV7. I have attached the corresponding files with the detected duplications labelled in colour. Please take a look and correct if needed. A brief explanation would be very helpful.
10. Our data editors have flagged the following issues in figure legends that need correcting:
 - Please provide the exact p values in the legends of figures 2c, e, g, j; 3a, h; 4b; 5c-e; EV 6a; EV 8b-c.
 - Please note that in figures 5c-e; there is a mismatch between the annotated p values in the figure legend and the annotated p values in the figure file that should be corrected.
 - Please define the scale bar for figure EV 1a.
 - Please define the white arrowheads in the legend of figure 2h-i.

Finally, I would like to suggest minor edits in the synopsis that you kindly provided. I have also written a short blurb that will accompany the title of your manuscript on our online page of contents. Please check the text below and let me know if any edits or corrections are needed:

Blurb:

Efficient neuronal migration requires activation of the mechanosensitive ion channel Tmem63b, and the resulting calcium influx and actomyosin contraction.

Synopsis:

Neurons migrate via repeated leading process extension and somal translocation. This study shows that these two steps are linked by mechanical signaling, induced by plasma membrane tension and mediated by mechanosensitive ion channels, allowing rapid and saltatory neuronal migration.

- Leading process extension increases plasma membrane tension in migrating neurons.
- The ensuing increase in tension activates the mechanosensitive ion channel Tmem63b and triggers Ca²⁺ influx.
- The resultant Ca²⁺ signaling induces actomyosin contraction and somal translocation.
- Tension-mediated signaling is required for efficient and saltatory 3D neuronal migration.

Furthermore, I would suggest adding a definition for Flipper-TR in the fifth sentence of the abstract:

"By using the membrane tension sensor Flipper-TR..."

With best wishes,

Ieva

We realize that it is difficult to revise to a specific deadline. In the interest of protecting the conceptual advance provided by the work, we recommend a revision within 3 months (6th Feb 2025). Please discuss the revision progress ahead of this time with the editor if you require more time to complete the revisions.

Referee #1:

I thank the authors for addressing all the issues that were raised.
I find it interesting that the calcium transients are sensitive to the direction of the stretch. The model of the authors does not predict a directional sensitivity (at least it is not clear to me how it could). The authors may want to add a comment about this issue. Otherwise I don't have any further comments.

Referee #2:

The authors did an excellent revision work and have answered all my concerns

Responses to the comment of Referee #1**(Manuscript Number: EMBOJ-2024-117904R).**

I find it interesting that the calcium transients are sensitive to the direction of the stretch. The model of the authors does not predict a directional sensitivity (at least it is not clear to me how it could). The authors may want to add a comment about this issue. Otherwise I don't have any further comments.

Response: We thank the referee for this comment. As described in the last revised manuscript, the perpendicular stretching stretches the growth cone and the soma. On the other hand, the parallel stretching stretches the leading process in addition to the growth cone and the soma. Indeed, the parallel stretching increased the leading process length (double arrow, **Data 1A for Referee #1**) and decreased leading process width (red arrowheads), and increased plasma membrane tension along the leading process shaft (Fig. 1). However, the perpendicular stretching did not affect both the leading process length and width (**Data 1B for Referee #1**), possibly due to a weaker interaction between the leading process shaft and the surrounding environment. Therefore, the perpendicular stretching cannot sufficiently increase the membrane tension along the neurite shaft. This is the reason why the calcium transients are sensitive to the direction of the stretch. In response to the referee's comment, we have clarified this point by adding an additional explanation in the revised manuscript (green font, line 128-129).

Dear Dr. Inagaki,

Thank you for addressing the final editorial points. I am now pleased to inform you that your manuscript has been accepted for publication in the EMBO Journal.

If you have any questions, please do not hesitate to contact the Editorial Office. Thank you for this contribution to The EMBO Journal and congratulations on a nice study!

Best wishes,

Ieva
